# Splicing factor BUD31 promotes ovarian cancer progression through sustaining the expression of anti-apoptotic *BCL2L12*

Zixiang Wang [1,2], Shourong Wang[1], Junchao Qin[1,2], Xiyu Zhang [1], Gang Lu[3], Hongbin Liu[4], Haiyang Guo[5], Ligang Wu [6], Victoria O. Shender [7], Changshun Shao [8] ✉, Beihua Kong [1] ✉ & Zhaojian Liu [1,2] ✉

Dysregulated expression of splicing factors has important roles in cancer development and progression. However, it remains a challenge to identify the cancer-specific splicing variants. Here we demonstrate that spliceosome component BUD31 is increased in ovarian cancer, and its higher expression predicts worse prognosis. We characterize the BUD31-binding motif and find that BUD31 preferentially binds exon-intron regions near splicing sites. Further analysis reveals that BUD31 inhibition results in extensive exon skipping and a reduced production of long isoforms containing full coding sequence. In particular, we identify *BCL2L12*, an anti-apoptotic BCL2 family member, as one of the functional splicing targets of BUD31. BUD31 stimulates the inclusion of exon 3 to generate full-length *BCL2L12* and promotes ovarian cancer progression. Knockdown of BUD31 or splice-switching antisense oligonucleotide treatment promotes exon 3 skipping and results in a truncated isoform of *BCL2L12* that undergoes nonsense-mediated mRNA decay, and the cells subsequently undergo apoptosis. Our findings reveal BUD31-regulated exon inclusion as a critical factor for ovarian cancer cell survival and cancer progression.

Ovarian cancer is the most lethal reproductive cancers, with over 184,000 deaths each year worldwide[1]. More than 75% of patients of ovarian cancer are diagnosed at late stage and has no effective screening strategy[2]. The standard treatment for advanced ovarian cancer is cytoreductive surgery followed by platinum-based chemotherapy. However, most patients will develop platinum-resistant disease[3]. Anti-angiogenics and PARP inhibitors have shown promising clinical results

for the treatment of ovarian cancer[4,5]. Recent studies show ATR inhibition overcomes PARP inhibitor and platinum resistance in ovarian cancer cells[6]. WEE1 inhibitor has been shown to sensitize chemotherapy particularly in TP-53 mutated ovarian cancer[7]. Despite these advances, relapse and chemotherapy resistance is a still major clinical problem.

Alternative splicing (AS) of precursor mRNA is an important step to increase the diversity of gene expression, and AS has been estimated

[1]Key Laboratory of Experimental Teratology, Ministry of Education, School of Basic Medical Science, Department of Obstetrics and Gynecology, Qilu Hospital, Cheeloo College of Medicine, Shandong University, Jinan, China. [2]Advanced Medical Research Institute, Meli lake Translational Research Park, Cheeloo College of Medicine, Shandong University, Jinan, China. [3]CUHK-SDU Joint Laboratory on Reproductive Genetics, School of Biomedical Sciences, The Chinese University of Hong Kong, Hong Kong, China. [4]Center for Reproductive Medicine, Cheeloo College of Medicine, Shandong University, Jinan, China. [5]Department of Clinical Laboratory, The Second Hospital of Shandong University, Cheeloo College of Medicine, Shandong University, Jinan, Shandong Province, China. [6]Shanghai Institute of Biochemistry and Cell Biology, Chinese Academy of Sciences, Shanghai, China. [7]Center for Precision Genome Editing and Genetic Technologies for Biomedicine, Federal Research and Clinical Center of Physical-Chemical Medicine of Federal Medical Biological Agency, 119435 Moscow, Russia. [8]Institutes for Translational Medicine, Soochow University, Suzhou, China. ✉e-mail: shaoc@suda.edu.cn; kongbeihua@sdu.edu.cn; liujian9782@sdu.edu.cn

to occur in about 95% of human multi-exon genes[8]. While AS is essential for normal development, dysregulation of the splicing process is also implicated in various diseases, including cancer[9], and malignant tumors have up to 30% more AS events than normal tissues[10]. AS is regulated by trans-splicing factors that specifically bind to cis-elements in pre-mRNAs[11].

Mutations or altered expression of splicing factors can lead to splicing reprogramming, which contributes to tumor initiation and progression. For example, SF3B1 mutations induce mis-splicing of *BRD9*, leading to its degradation and the promotion of melanomagenesis[12]. U2AF1 mutations cause abnormal recognition of the 3′ splice site of pre-mRNA, resulting in increased DNA damage in cancers[13]. SRSF1 is overexpressed in various cancers and exerts oncogenic roles by regulating the AS of genes, including MYO1B[14]. The splicing factor ESRP1 regulates *CD44* splice switching during the epithelial-mesenchymal transition[15], and SF3B2 drives prostate cancer progression through AS of androgen receptor (AR) to increase AR-V7 expression[16]. Conversely, RBM4 is downregulated in cancer tissues, and this suppresses tumor progression by modulating *Bcl-x* splicing[17]. Dysregulation of AS provides a promising therapeutic strategy for cancer treatment. For example, H3B-8800, an SF3B1 inhibitor, is currently in a phase I trial for the treatment of various cancers[18], and antisense oligonucleotides (ASOs) for modulating pre-mRNA splicing are currently FDA-approved for the treatment of spinal muscular atrophy[19]. ASOs modulate *Bcl-x* splicing to produce the pro-apoptotic isoform and reduce the tumor load in mice[20], suggesting a promising approach for cancer therapy. Multiple AS markers have been identified in ovarian cancer[21], including aberrant expression of splicing factors such as SRSF3 and SFPQ[22,23]. Hyperactivation of MYC can lead to global upregulation of pre-mRNA levels and to aberrant splicing patterns in ovarian cancer[24]. However, knowledge of the role of splicing factors in the generation of ovarian cancer-related splicing variations is still limited.

BUD31 is a spliceosomal component in yeast and is required for spliceosome assembly and catalytic activity[25]. BUD31 is identified as a MYC-synthetic lethal gene in human mammary epithelial cells[26], implying its potential role in cancer. Nonetheless, the alternative splicing regulation and clinical significance of BUD31 in cancer remain poorly understood. In this work, we report that overexpression of BUD31 predicts poor prognosis in ovarian cancer patients. Furthermore, we have used RNA-seq and CLIP-seq analysis to show BUD31-regulated AS and identify the binding motif and preferred genome-wide binding pattern of BUD31. More importantly, we verify that BUD31 drives an oncogenic splicing switch of *BCL2L12*, which in turn promotes ovarian cancer progression. Our study indicates that BUD31 is a critical oncogenic splicing factor that might act as a potential therapeutic target in ovarian cancer.

## Results

### Elevated BUD31 expression is associated with poor prognosis in ovarian cancer

To identify survival-related splicing factors in serous ovarian cancer (SOC), we first analyzed the expression of 134 known splicing factors[27] in SOC tissues ($n = 374$) compared to normal tissues ($n = 180$) from the TCGA and GTEx databases. We identified 20 upregulated and 17 downregulated splicing factors in SOCs (Fig. 1a and Supplementary Data 1). We next assessed the prognostic values of 37 dysregulated splicing factors in patients with SOC and found 10 of 37 splicing factors to be significantly related to both progression-free survival and overall survival (Fig. 1b, c). We then conducted a RNAi screen of six splicing factors and found BUD31 and SF3B1 exhibited similar inhibitory effect on ovarian cancer cells (Figs. 1d, e and S1a−c). Considering that high BUD31 expression predicts worse prognosis in ovarian cancer, we chose BUD31 for further investigation. We examined the expression level of BUD31 in the TCGA and Clinical Proteomic Tumor Analysis

Consortium (CPTAC) data. BUD31 was commonly upregulated in SOC compared with normal samples at both the RNA and protein levels (Fig. 1f, g). Importantly, BUD31 protein level was significantly increased in advanced ovarian cancer compared to early-stage patients (Fig. 1g). Pan-cancer analysis revealed that BUD31 is overexpressed in various cancer types (Fig. S1d). To evaluate the clinical significance of BUD31 in SOC, we performed immunohistochemistry using a tissue microarray containing 149 ovarian cancer tissue samples and 73 fallopian tube tissues (FTs). Immunohistochemical (IHC) staining revealed the significant overexpression of BUD31 in SOCs (Fig. 1h, i). We further assessed the prognostic value of BUD31 in SOCs in this cohort and found that a high level of BUD31 was significantly correlated with poor overall survival and progression-free survival in patients with SOC (Fig. 1j, k). This correlation was also verified by the Kaplan–Meier Plotter database based on RNA expression data (Fig. 1l, m). Moreover, a high level of BUD31 was positively associated with being younger than 60 years old (Supplementary Table 1). Together, these results demonstrate that elevated expression of BUD31 is associated with worse prognosis in ovarian cancer patients.

### Knockdown of BUD31 induces spontaneous apoptosis in ovarian cancer cells

To investigate the functional role of BUD31 in ovarian cancer, we established cell lines with BUD31 overexpression or BUD31 knockdown relative to the basal expression level of BUD31 (Fig. S1e). We then performed RNA-seq on BUD31 knockdown and control HEYA8 cells and identified 1,243 downregulated and 943 upregulated genes (Fig. S2a, b). Among the downregulated genes upon BUD31 knockdown, 31.86% were oncogenic genes highly expressed in SOCs (Fig. S2c). Gene Ontology (GO) analysis revealed that BUD31 target genes were enriched in biological processes including apoptosis, cell division, and microtubule cytoskeleton organization (Fig. 2a, b). Consistent with this, the gene set enrichment analysis (GSEA) demonstrated that BUD31 knockdown regulates the apoptosis signaling pathway (Fig. 2c). We next measured apoptosis in ovarian cancer cells upon BUD31 knockdown using Annexin V-PE/7-AAD staining and flow cytometry. As expected, knockdown of BUD31 induced significant spontaneous apoptosis in HEYA8 and OV90 cells (Fig. 2d). Consistent with this, overexpression of BUD31 suppressed $H_2O_2$-induced ovarian cancer cell apoptosis (Fig. 2e). The expression of apoptosis-related proteins was next measured by western blot, and inactivation of BUD31 resulted in an increased level of cleaved caspase-3 and PARP1, whereas overexpression of BUD31 had opposite effects (Figs. 2f and S2d, e). Additionally, BUD31 knockdown using siRNAs resulted in depolymerized microtubules and abnormal cell morphology in HEYA8 cells (Fig. S2f, g). These results indicate that high levels of BUD31 exert an anti-apoptosis effect in ovarian cancer.

### BUD31 promotes proliferation and xenograft tumor growth in ovarian cancer

To further explore the function of BUD31 in ovarian cancer, we performed an EdU (5-ethynyl-2′-deoxyuridine) assay and found that BUD31 overexpression in HEYA8, A2780 and OVCAR3 cells significantly increased the number of EdU-positive cells. In contrast, knockdown of BUD31 in HEYA8, OV90, OVBWZX (primary ovarian cancer cells derived from the ascites of SOC patients, verified by PAX8 and p53) reduced the number of EdU-positive cells (Figs. 3a and S3a, b). Additionally, growth curve and clonogenic assays showed that BUD31 overexpression significantly enhanced the proliferation of ovarian cancer cells, while silencing BUD31 had opposite effects (Figs. 3b and S3c, d). Moreover, mouse xenograft experiments were conducted to assess the functional role of BUD31 in ovarian cancer tumorigenesis and progression. Luciferase-expressing HEYA8 cells with dox-inducible BUD31 knockdown and corresponding control cells were intraperitoneally injected into nude mice ($n = 6$), and luciferase was

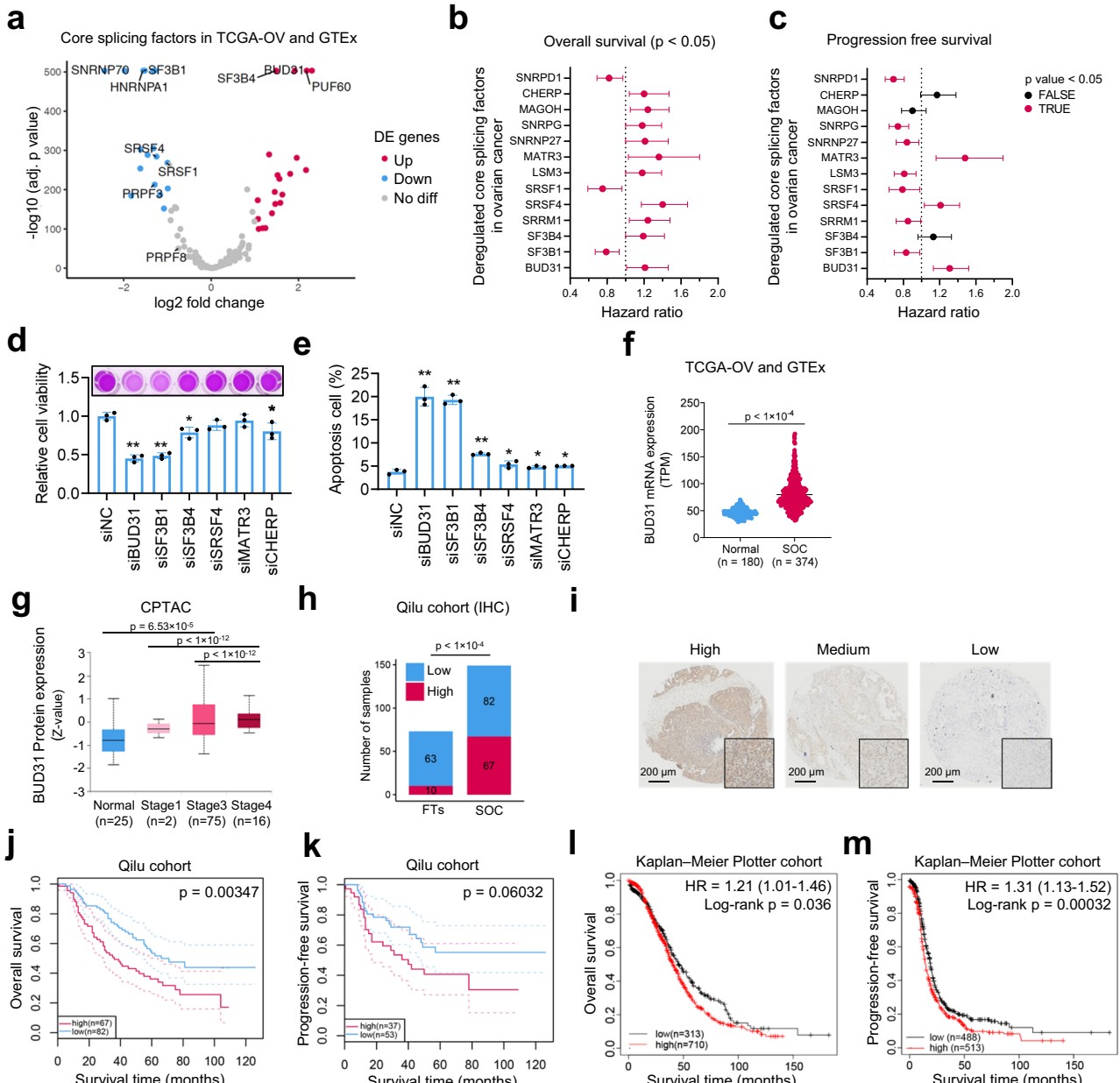

**Fig. 1 | Elevated BUD31 expression is associated with poor prognosis in ovarian cancer. a** Volcano plot of differentially expressed core splicing factors (*n* = 134) between the TCGA-OV cohort (*n* = 374) and normal tissue in GTEx datasets (*n* = 180). |log2FC| > 1 and an adjusted *p* value < 0.05 were considered significant. **b, c** Forrest plot of the hazard ratio for the association between 13 dysregulated splicing factors and overall survival (*n* = 1023) and progression-free survival (*n* = 1001) in patients with SOC from the Kaplan−Meier Plotter database. The cohort of patients with SOC was split by auto-select cutoff. Prognostic splicing factors (red) with a *p* < 0.05 were considered statistically significant. **d, e** MTT assay and flow cytometry determined the HEYA8 cell viability and apoptosis cell percentage after siBUD31, siSF3B1, siSF3B4, siSRSF4, siMATR3, and siCHERP treatment for 72 h (*n* = 3 biologically independent experiments). *\*p* < 0.05, **\*\*p* < 0.01. **f** *BUD31* mRNA expression was analyzed in the TCGA-OV cohort (*n* = 374) and normal tissue in GTEx datasets (*n* = 180). **g** BUD31 protein level was analyzed in SOCs (*n* = 93) and FTs (*n* = 25) from the CPTAC dataset. SOC samples were classified into Stage 1

(*n* = 2), Stage 3 (*n* = 75), and Stage 4 (*n* = 16) according to individual cancer stage. The bounds of the box were the upper and lower quartile with the median value in the center. The whiskers indicated the minima and maxima. **h** Statistical analysis of BUD31 expression from IHC staining of the tissue microarray containing 149 samples of SOCs and 73 samples of FTs. **i** Representative images of IHC staining with high, medium, and low BUD31 expression in our tissue microarray. **j, k** Kaplan−Meier analysis of the correlation between BUD31 expression and overall survival and progression-free survival of ovarian cancer patients based on data from our tissue microarray. The 95% confidence interval was shown as dotted lines. **l, m** Kaplan−Meier analysis of the correlation between BUD31 expression and overall survival and progression-free survival of ovarian cancer patients based on the Kaplan−Meier Plotter cohort. The *p* value was obtained by log-rank test (**b, c, j, k, l, m**), two-tailed unpaired Student's *t*-test (**d, e, f, g**), and Chi-square test (**h**). Data are presented as means ± SD unless otherwise stated.

used as a tracer for in vivo imaging analysis. BUD31 knockdown in HEYA8 cells significantly reduced the number and size of tumor nodes in the abdominal cavity (Figs. 3c and S3e, f). We also found that BUD31 knockdown reduced the Ki-67 index (Fig. 3d). Furthermore, BUD31 knockdown enhanced apoptosis in xenograft tumors of HEYA8 cells as

detected by TUNEL assay (Fig. 3e). Consistent with this, forced expression of BUD31 in ID8 cells induced a significant increase in tumor mass and volume in xenografts (*n* = 8) (Fig. S3g−i). Therefore, these results suggest that BUD31 exhibits oncogenic potential in ovarian cancer.

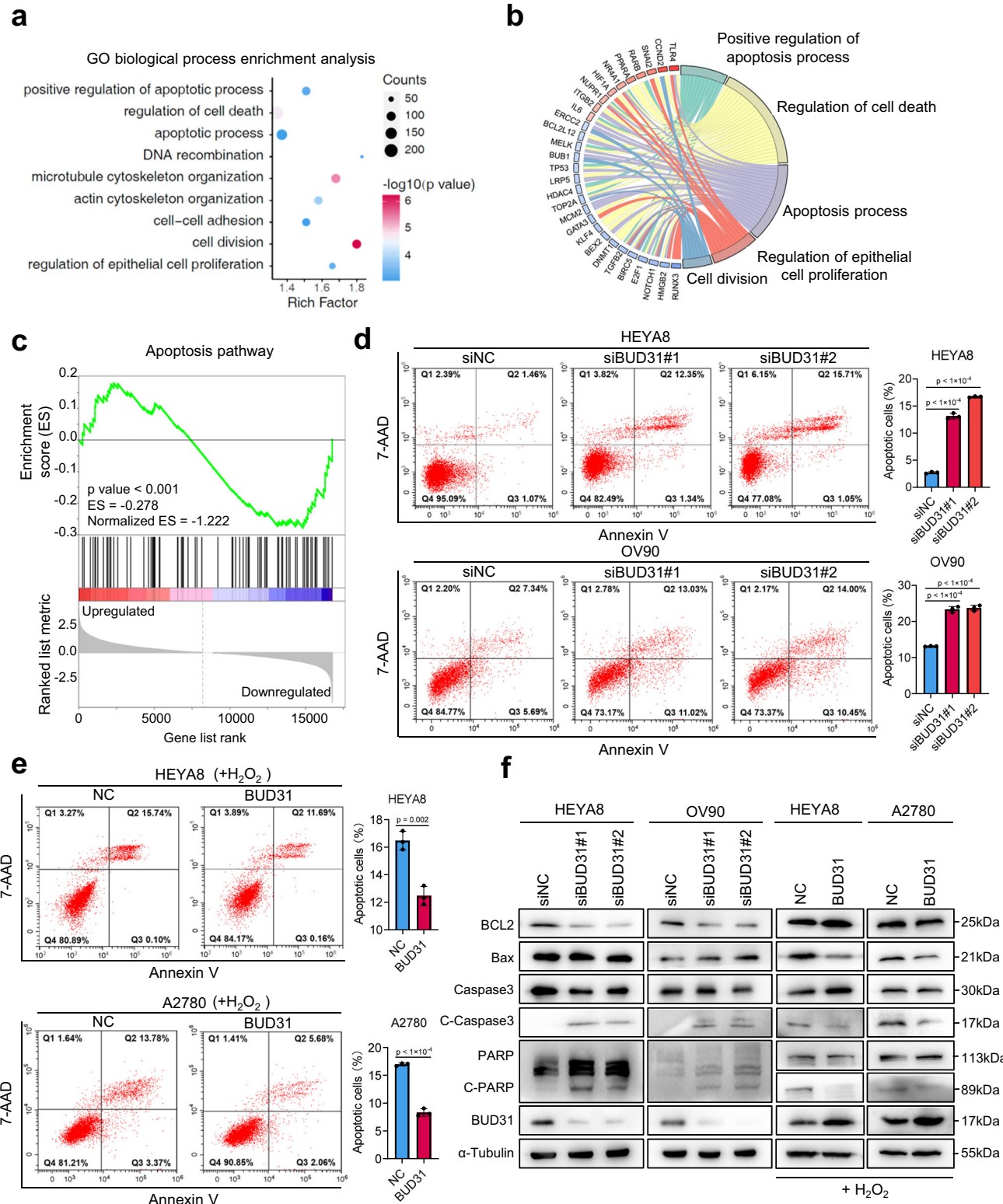

**Fig. 2 | Knockdown of BUD31 induces spontaneous apoptosis in ovarian cancer cells.** GO biological process enrichment (**a**) and Circos plot (**b**) analysis were conducted on DEGs in the RNA-seq data from HEYA8 cells after BUD31 knockdown. **c** GSEA analysis was performed with the gene expression profile after BUD31 knockdown (Normalized ES = −1.222, $p < 0.001$). **d, e** Apoptotic cells were detected by flow cytometry after staining with Annexin V-PE/7-AAD in ovarian cancer cells after BUD31 knockdown with two strands of siRNAs (HEYA8 and OV90) or overexpression (HEYA8 and A2780) ($n = 3$ biologically independent experiments).

Apoptotic cells percentage included early and late apoptotic cells. Cells overexpressing BUD31 were treated with $H_2O_2$ with a final concentration of 400 μM for 4 h before apoptosis detection. **f** Apoptotic markers were measured by western blot. BUD31 was knocked down in HEYA8 and OV90 cells and was overexpressed HEYA8 and A2780 cells in the presence of $H_2O_2$. The $p$ value was obtained by two-tailed unpaired Student's $t$-test (**e, f**), and data are presented as means ± SD. Source data are provided as a Source Data file.

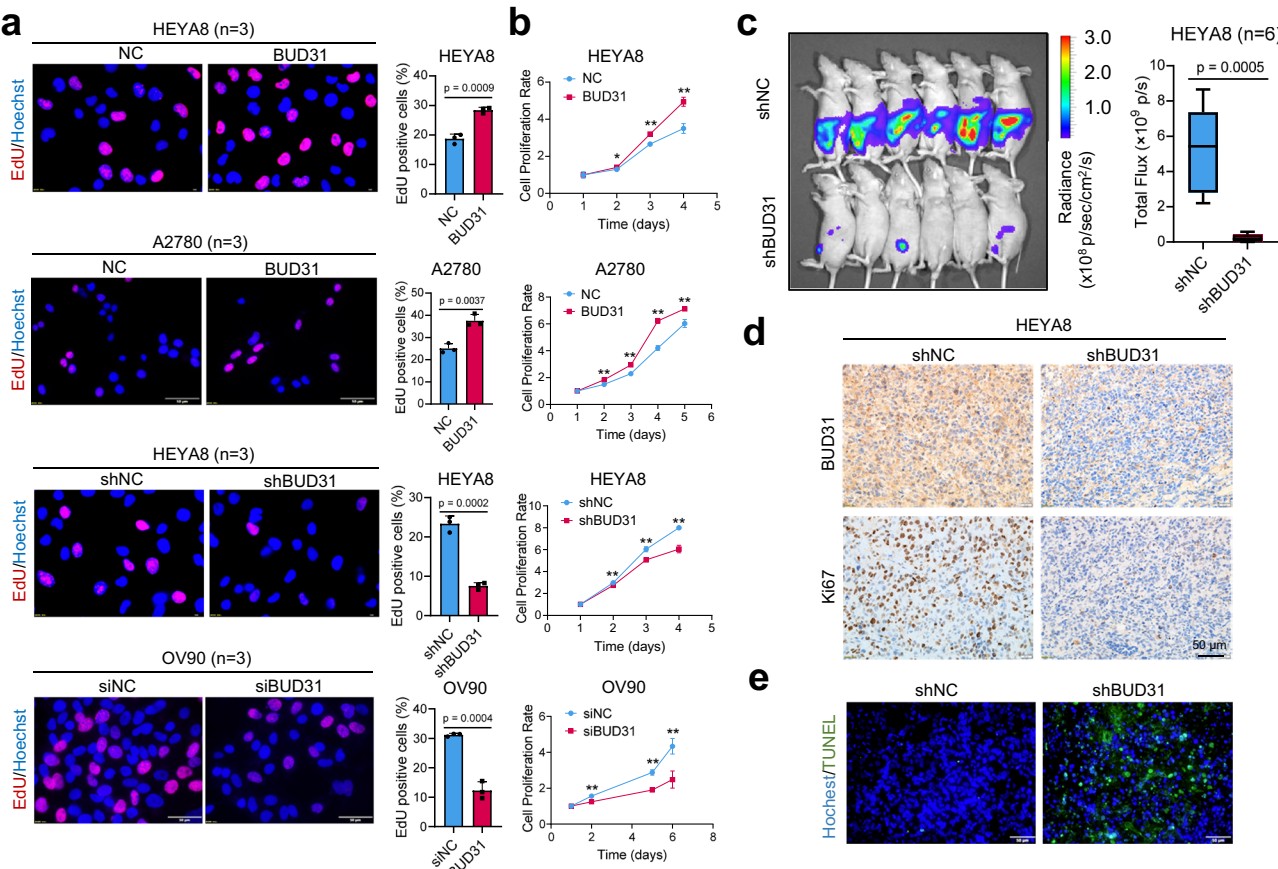

**Fig. 3 | BUD31 promotes proliferation and xenograft tumor growth in ovarian cancer.** The EdU assay (**a**) and cell proliferation assay (**b**) were performed in ovarian cancer cells with BUD31 overexpression (HEYA8, A2780) or knockdown (HEYA8, OV90) compared to corresponding controls ($n = 3$ for the EdU and $n = 5$ biologically independent experiments for the cell proliferation assay). Cell proliferation was measured using the MTT cell proliferation assay. Absorbance at 570 nm at each time point was compared to the initial state (time = 1 day). **c** Luciferase signals of intraperitoneal injected nude mice and photon flux quantification. Nude mice were injected with luciferase-expressing HEYA8 cells with a dox-inducible BUD31 knockdown system ($n = 6$ per group). Administration of doxycycline (1.2 g/L) started one week after the cell implantation. The bounds of the box were the upper and lower quartile with the median value in the center. The whiskers indicated the minima and maxima. **d** IHC staining of BUD31 and Ki-67 expression in xenograft tumors of HEYA8 cells with BUD31 knockdown compared to corresponding controls. **e** TUNEL assay to quantify the apoptotic cells in xenograft tumors with BUD31 knockdown compared to corresponding controls. The $p$ value was obtained by two-tailed unpaired (**a**–**c**), and the results are presented as the mean ± SD. *$p < 0.05$, **$p < 0.01$. Source data are provided as a Source Data file.

## Identification of the genome-wide landscape of BUD31-binding sites on RNA

To unveil the role of BUD31 in AS, we first identified proteins that are associated with BUD31 by immunoprecipitation with BUD31 antibody coupled to mass spectrometry (IP-MS) in HEYA8 cells. GO enrichment analysis showed that the mRNA splicing via the spliceosome and regulation of RNA splicing pathways were significantly enriched among proteins interacting with BUD31 (Fig. 4a). Intriguingly, we found that 46 annotated spliceosome proteins were associated with BUD31 (Fig. S4a and Supplementary Data 2). Among these, BUD31 immunoprecipitated predominantly with U2 snRNPs and hnRNP proteins (Fig. 4b). U2AF1, HNRNPU, SNRPA1, and SART1 were verified as BUD31-interacting partners as determined by co-immunoprecipitation assay (Fig. S4b). Interestingly, immuno-fluorescence assay showed that BUD31 was colocalized with SC35, which is a marker of the nuclear speckle (a type of nuclear body involved in splicing factor storage) (Fig. 4c). The interactions between BUD31 and multiple spliceosome components suggest an essential role for BUD31 in the regulation of AS.

To generate genome-wide maps of BUD31 protein-RNA interactions, we performed SpyTag-based CLIP (SpyCLIP) for BUD31 in HEYA8 cells as previously described[28]. SpyCLIP is a covalent link-based CLIP method with high efficiency and accuracy. SpyTag-

SpyCatcher system could withstand harsh washing for removing non-specific interactions[29]. The SpyCLIP-seq data with high accuracy and reproductivity was suitable for further analysis (Fig S4c–i). A total of 9,983,683 reads in the input group and 30,970,512 reads in the SpyCLIP group were obtained, and 99.02% of the SpyCLIP reads were mapped to an annotated human genome (hg38). Further cluster analysis revealed that BUD31 binds to 8780 annotated human genes. Protein-RNA crosslink sites were identified by the PURECLIP method, which explicitly incorporates CLIP truncation patterns and non-specific sequence biases[30]. The identified regions shared more SpyCLIP reads than the control group, and SpyCLIP exhibited strong enrichment at the crosslink sites (Figs. 4d, e and S4j). Most BUD31-binding regions were less than 280 nucleotides in length and contained less than six crosslink sites (Fig. 4f), and more than 87.60% of the BUD31-RNA crosslink regions mapped to exons and introns (Fig. 4g, h). To better characterize the interaction of BUD31 with RNA, we used the HOMER algorithm[31] to identify the BUD31-recognizing RNA motif and found that the most abundant element was the ACUUACCU 8-mer (Fig. 4i–k). Strikingly, 2 of the 4 top-scoring motifs (motif 1 and motif 3) were located near the 5ss intron-exon junction and were reverse complemented. The other two top-scoring motifs were located in exon regions (Figs. 4l and S4k). We further conducted a correlation analysis between BUD31-

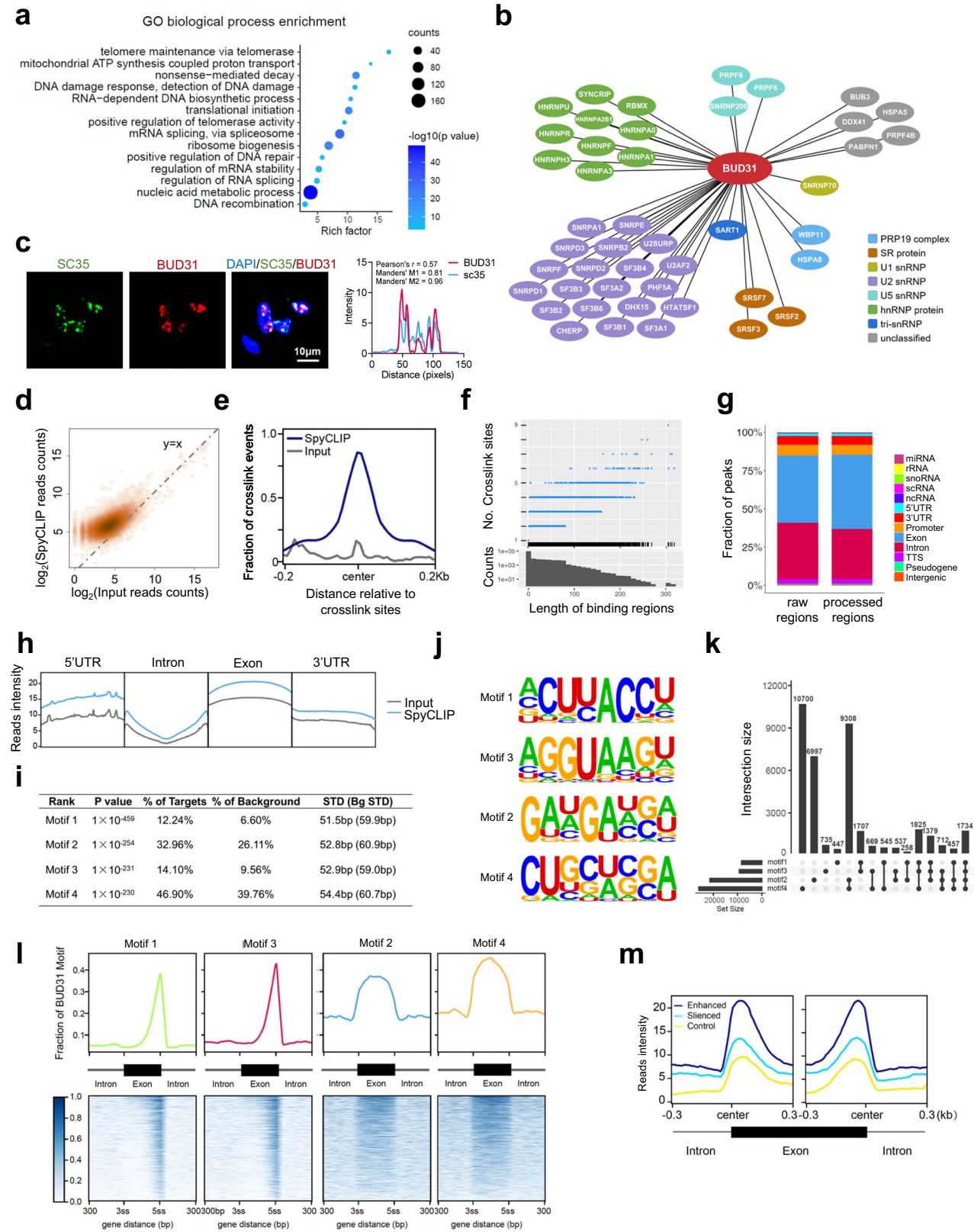

binding regions and the regulated alternative exons based on Spy-CLIP and RNA-Seq data. Intriguingly, the BUD31-binding sites were highly enriched in exon-intron regions near both the 3' and 5' splicing sites (Figs. 4m and S4l). To sum up, these results suggest that BUD31 exerts its function in AS through direct interactions with the pre-mRNA substrate.

## Global identification of AS events regulated by BUD31

To investigate BUD31-regulated AS events, AS analysis was performed based on the RNA-seq data in BUD31 knockdown and control HEYA8 cells. Differential AS events including retained introns, skipped exons, alternative 5' splice sites, alternative 3' splice sites, and mutually exclusive exons were identified. The predominant AS event upon

**Fig. 4 | Identification of the genome-wide BUD31-binding sites on RNA. a** GO biological process enrichment analysis of BUD31-interacting proteins in HEYA8 cells using immunoprecipitation coupled to mass spectrometry. **b** The correlation network between BUD31 and splicing factors was constructed in Cytoscape. Proteins belonging to the spliceosome were classified into PRP19 complex, SR protein, U1 snRNP, U2 snRNP, U5 snRNP, hnRNP protein, tri-snRNP, and others. **c** Immunofluorescence assays showed the co-localization of BUD31 (red) with splicing factor SC35 (green) in punctate nuclear speckles. The quantification and analysis of co-localization was performed with Coloc 2 and Plot Profile (Pearson's $r = 0.57$, Manders' M1 = 0.81, Manders' M2 = 0.96). **d** High-density scatter plot of the SpyCLIP and Input reads counts aligned to the BUD31-binding regions. **e** Position of the SpyCLIP crosslinking regions relative to the crosslinking sites identified by PURECLIP. The SpyCLIP (deep blue) and input (gray) signals are shown around the crosslink sites. **f** Length distribution of the BUD31-binding regions and the crosslinking sites included in the corresponding binding regions. **g** Distribution of SpyCLIP crosslinking regions annotated by HOMER on genome elements. Processed regions were longer than 3 nucleotides. **h** SpyCLIP read distribution compared with input on the genome elements, including 5' UTR, intron, exon, and 3' UTR. **i** De novo motif analysis of BUD31 SpyCLIP clusters and statistical results of the top-four BUD31-binding motifs ranked by HOMER calculated $p$ value. **j** Enriched sequence elements of the top-four BUD31-binding motifs. **k** Upset plot of the distribution of motifs 1–4 in the BUD31-binding regions. **l** BUD31-binding motif distribution in the exon region and the 300 bp flanking the 3ss or 5ss intron-exon junction site. The exon region was scaled such that the length was equal to 300 bp to normalize different exon lengths. **m** SpyCLIP reads intensity distributed around the alternative exons. Enhanced exons (deep blue) were included, and silenced exons (blue) were excluded after silencing BUD31. Randomly chosen exons were used as controls (yellow).

BUD31 knockdown was skipped exons (68.0%), the percentage decreased to 57% in RNA-seq data of HEYA8 cells with BUD31 over-expression (Fig. 5a, b), suggesting that BUD31 promotes exon inclusion. Similar proportions of exon skipping were observed in A2780 and SKOV3 cells upon BUD31 knockdown (Fig. S5a–c). Further global AS analysis revealed that BUD31 knockdown increased skipped exons and retained introns, BUD31 overexpression had the opposite effects (Fig. 5c, d). Consistent with this, calculation of the global coding sequence length (CSL) showed that knockdown of BUD31 significantly decreased the abundance of long CSL isoforms (750–1750 bp) and increased the abundance of short CSL isoforms (100 bp to 650 bp) (Figs. 5e and S5d). Thus, the average CSL of all isoforms decreased after BUD31 silencing (Fig. 5f). Additionally, 3'UTR length increased, whereas 5'UTR length was not affected upon BUD31 knockdown (Fig. S5e, f). Moreover, we analyzed the nonsense-mediated mRNA decay (NMD) sensitive splice isoforms of BUD31-regulated AS because coupled AS and NMD might regulate many genes[32]. A total of 6325 NMD-sensitive splice isoforms were identified, and the expression level of the genes that acquired increased NMD-sensitive isoform fraction decreased upon BUD31 knockdown (Fig. 5g). The NMD-sensitive targets were enriched in DNA repair, mitotic cell cycle process, DNA replication, and apoptotic signaling pathways (Fig. 5h).

In order to identify functional target candidates involved in ovarian cancer progression, we integrated BUD31-bound genes and AS-related genes using CLIP-seq, RIP-seq and RNA-seq data. Combined analysis revealed that 317 genes with AS events upon BUD31 knock-down were also bound by BUD31 (Fig. 5i and Supplementary Data 3). Mitotic cell cycle related targets E2F4 and CDK16, among others, were successfully validated by semi-quantitative RT-PCR (Fig. 5j–n). Intron 2 of E2F4 and intron 12 of CDK16 were more likely to be retained due to BUD31 ablation, which reduced the subsequent protein expression (Fig. 5l). Moreover, to obtain more specific results, Spliceseq, MISO and rMATS algorithm were used in AS detection and 105 differential AS events were identified (Fig. S5g, h) and 22 of them were bound with BUD31 (CLIP-seq and RIP-seq). Function annotation revealed that AS candidates BCL2L12 and RBCK1 were responsible for apoptotic signaling pathways and BCL2L12 possessed a stronger anti-apoptosis and proliferation ability than RBCK1 (Fig. 5o–r). Collectively, these data imply that BUD31 is a functional regulator of AS and predominantly regulates exon skipping and intron retention.

## BUD31 promotes *BCL2L12* exon 3 inclusion through direct binding to the pre-mRNA

Of the BUD31-bound and alternatively spliced transcripts, BCL2L12, an anti-apoptotic BCL2 family member, caught our attention. BCL2L12 is overexpressed in glioblastoma and has been identified as a rational therapeutic target in glioblastoma[33]. Recently, therapeutic RNAi targeting *BCL2L12* has been conducted in a first-in-human trial in glioblastoma[34]. We first analyzed RNA-seq data and found BUD31 knockdown promoted exon 3 skipping and resulted in a short isoform

(*BCL2L12*-S). Strikingly, BUD31 was shown to bind to exon 3 of *BCL2L12* according to the CLIP-seq and RIP-seq data, indicating that *BCL2L12* is a direct target of BUD31 (Fig. 6a, b). We next verified the role of BUD31 in the AS of *BCL2L12* by semi-quantitative RT-PCR and fragment analysis. We found that BUD31 knockdown promoted exon 3 skipping to generate more *BCL2L12*-S but less full-length isoform (*BCL2L12*-L), overexpression of BUD31 had an opposite effect (Figs. 6c, d and S6a, b). We further determined the *BCL2L12*-L/*BCL2L12*-S ratio using isoform-specific primers. Importantly, knockdown of BUD31 in HEYA8 cells significantly decreased the *BCL2L12*-L/*BCL2L12*-S ratio, whereas ectopic expression of BUD31 had the opposite effect (Fig. 6e). Furthermore, sequence analysis showed that the exon 3 skipping of *BCL2L12* caused a frameshift that introduced a premature termination codon (Fig. 6f), and transcripts with such termination codons are predicted to be degraded by NMD[35]. To verify that *BCL2L12*-S is degraded by NMD pathway, we measured the RNA half-life of *BCL2L12*-S in UPF1 knock-down and control HEYA8 cells treated with the transcription inhibitor actinomycin D. Notably, the half-life of *BCL2L12*-S was significantly increased and the *BCL2L12*-S/*BCL2L12*-L ratio was higher in UPF1 knockdown cells relative to controls, confirming that *BCL2L12* is sensitive to NMD (Figs. 6g, and S6c, d). To obtain further evidence for the direct binding of BUD31 to *BCL2L12* pre-mRNA, we performed an RNA immunoprecipitation (RIP) assay in HEYA8 cells with BUD31 antibody or control IgG. RIP-qPCR showed that BUD31 bound to exon 2, exon 3, and intron 3, but not to exon 1 or intron 2 (Fig. 6h). An RNA pull-down assay further revealed that BUD31 was more abundantly enriched by wildtype probe, but not by mutant probe spanning exon 3 to intron 3 (Figs. 6i and S7e, f). Moreover, an RNA electrophoretic mobility shift assay (EMSA) confirmed the interaction between BUD31 and *BCL2L12* pre-mRNA spanning exon 3 to intron 3 (Fig. 6j). These findings strongly support the notion that BUD31 promotes *BCL2L12* AS through direct binding to the pre-mRNA.

To determine the clinical relevance of *BCL2L12* exon 3 skipping in ovarian cancer, we analyzed the expression of *BCL2L12*-L (exon 3 inclusion) and *BCL2L12*-S (exon 3 skipping) in the TCGA ovarian cancer database and found that *BCL2L12*-L was significantly increased whereas *BCL2L12*-S was decreased in SOC (Fig. 6k, l). In addition, BUD31 was positively correlated with plenty of inclusion events, including *BCL2L12* AS, in ovarian cancer based on PSI value profiles in the TCGA-Spliceseq database (Fig. 6m). More importantly, ovarian cancer patients with exon 3 inclusion in the TCGA-OV cohort showed poor overall survival (Fig. 6n), implying that inclusion of *BCL2L12* exon 3 may be involved in ovarian cancer progression.

## The oncogenic roles of BUD31 in ovarian cancer partially rely on *BCL2L12* expression

To investigate whether the regulation of *BCL2L12* splicing by BUD31 mediate the oncogenic role of BUD31 in ovarian cancer, we first analyzed TCGA data and found that BCL2L12 was positively correlated with BUD31 (Fig. 7a). Next, we measured the protein level of

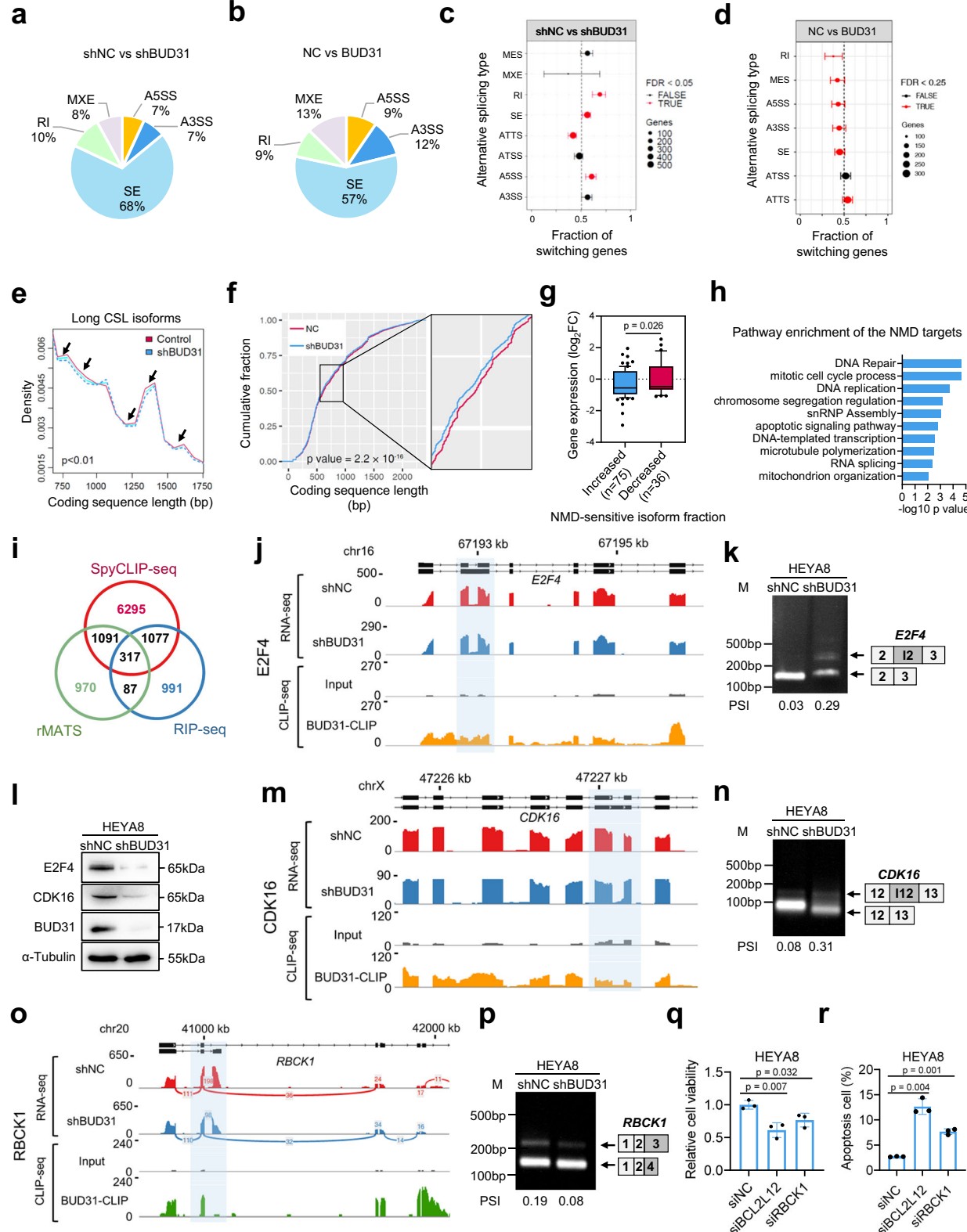

BCL2L12 in ovarian cancer cells with BUD31 knockdown or overexpression. BUD31 knockdown decreased the protein level of BCL2L12 while forced expression of BUD31 had the opposite effect (Fig. 7b). In addition, silencing BUD31 also decreased BCL2L12 expression in the xenograft tumor in vivo (Fig. S6g). We then conducted rescue experiments to determine whether the phenotype induced by BUD31 knockdown could be rescued by BCL2L12 overexpression. Indeed, BUD31 knockdown-induced apoptosis was

effectively attenuated by BCL2L12 overexpression (Fig. 7c, d). Additionally, the upregulation of BCL2L12 significantly reversed the inhibition of proliferation and clonogenic ability by BUD31 knockdown in HEYA8 cells (Fig. 7e, f). Moreover, overexpression of BCL2L12 rescued the inhibition of xenograft growth induced by silencing BUD31 (Fig. 7g–i). In contrast, overexpression of BUD31 decreased $H_2O_2$-induced apoptosis, and this could be reversed by BCL2L12 knockdown in A2780 cells (Fig. S7a, b). Meanwhile,

**Fig. 5 | Global identification of AS events regulated by BUD31. a, b** Pie chart depicting the proportions of different types of AS events in the RNA-seq data from HEYA8 cells after BUD31 knockdown or overexpression. SE skipped exons, RI retained introns, A5SS alternative 5′ splice site, A3SS alternative 3′ splice site, MXE mutually exclusive exons. **c, d** The proportions of genes (95% confidence interval) with different types of AS induced isoform switches after BUD31 knockdown ($n = 5651$) and overexpression ($n = 2723$). **e** Density plot of the long coding sequence isoforms (750–1750 bp) length distribution in HEYA8 cells with BUD31 knockdown compared with corresponding controls. Arrow, statistically significant regions. **f** Cumulative distribution function plot of coding sequence lengths of all annotated genes after BUD31 knockdown. **g** The differential expression of genes with increased NMD-sensitive isoforms fraction (dIF > 10%, $n = 75$) and decreased fraction (dIF < −10%, $n = 36$). The $\log_2$ transformed fold changes were shown in boxplot (10-90 percentile). **h** Pathway enrichment of the NMD-sensitive and downregulated targets. **i** Venn diagram of 8780 BUD31-binding genes from the SpyCLIP-seq, 2472 genes from RIP-seq, and 2465 genes with AS events. The AS pattern and BUD31-binding sites in *E2F4* (**j**), *CDK16* (**m**) and *RBCK1* (**o**) were visualized with IGV using the RNA-seq and SpyCLIP-seq data. The light blue region highlights the AS region and the BUD31-binding sites. Semi-quantitative RT-PCR was performed to validate the AS events in *E2F4* (**k**), *CDK16* (**n**), and *RBCK1* (**p**) in HEYA8 cells with BUD31 knockdown compared with controls. **l** The protein expression of E2F4 and CDK16 was determined by western blot in HEYA8 cells knocking down BUD31. **q, r** MTT assay and flow cytometry were performed to determine the cell viability and apoptosis cells percentage after siBCL2L12 and siRBCK1 treatment. The $p$ value was obtained by two-tailed unpaired Kolmogorov–Smirnov test (**f**) and Student's $t$-test (**g, q, r**), and the results are presented as the mean ± SD.

BCL2L12 knockdown significantly suppressed the BUD31-induced proliferation and clonogenic ability of A2780 cells (Fig. S7c, d). Importantly, pan-cancer analysis of the TCGA data showed that *BCL2L12* expression level and the proportion of *BCL2L12*-L were higher in most types of tumors than in corresponding normal tissues (Fig. S7e, f). Especially, BCL2L12 was overexpressed in SOCs, and high BCL2L12 expression levels predicted a poor prognosis for patients with SOC (Fig. 7j–l). Interestingly, knockdown of BCL2L12 significantly led to a reduction in BCL2 protein level, whereas there was no significant change of BCL2L12 after BCL2 knockdown (Fig. S7g–i), indicating that BCL2L12 may act upstream in the BCL2 apoptotic pathway. These results suggest that BUD31 exerts its oncogenic roles in ovarian cancer partially through regulating *BCL2L12*.

### ASO-mediated *BCL2L12* exon skipping induces apoptosis of ovarian cancer cells

Splice-switch ASOs are a promising strategy for the treatment of various diseases, including cancer, and ASOs have been approved for the treatment of spinal muscular atrophy and Duchenne muscular dystrophy[36]. To identify effective ASOs that promote *BCL2L12* exon 3 skipping, we analyzed the BUD31-binding region on *BCL2L12* based on CLIP-seq and designed three ASOs modified with phosphorothioate linkages (Fig. 8a). We examined the endogenous *BCL2L12* splicing in A2780 cells transfected with ASOs and found that ASO2 and ASO3 efficiently reduced *BCL2L12*-L (exon 3 inclusion) and increased *BCL2L12*-S (exon 3 skipping) levels when compared to control (Fig. 8b). Consistently, BCL2L12 protein expression was significantly decreased upon ASO2 and ASO3 treatment (Figs. 8c and S8a). ASO2 was chosen for subsequent experiments because it had the strongest effect on exon 3 skipping of *BCL2L12*. ASO2 was transfected into HEYA8 and A2780 cells and splicing of exon 3 was measured by RT-PCR. A dose-dependent increase in exon 3 skipping was observed after ASO2 treatment (Fig. 8d, e). Correspondingly, ASO2 decreased BCL2L12 expression at protein level in a dose and time-dependent manner (Fig. 8f, g). Importantly, ASO2 treatment induced apoptosis of A2780 and HEYA8 cells as determined by flow cytometry and cleaved-caspase-3 expression (Figs. 8f–h and S8b). Meanwhile, EdU incorporation assay showed that ASO2 significantly suppressed the proliferation of ovarian cancer cells (Fig. 8i). We further determined the half-maximal inhibitory concentration (IC50) values for ASO2 to be 74.27 nM and 73.70 nM in A2780 and HEYA8 cells, respectively (Fig. 8j). Moreover, a subcutaneous xenograft model established using HEYA8 cells showed that ASO2 treatment significantly reduced the tumor size and increased apoptosis (Fig. 8k–n). Interestingly, the spliceosome inhibitors pladienolides B and isoginkgetin could also induce *BCL2L12* exon 3 skipping and inhibit the proliferation of ovarian cancer cells (Fig. S8c–f). Taken together, these results suggest that ASO2 inhibits ovarian cancer cells proliferation by regulating *BCL2L12* exon 3 skipping.

## Discussion

Dysregulated expression of splicing factors and perturbed splicing have been shown to drive oncogenesis. We carried out a screen for survival-related splicing factors in SOC using TCGA data. We found that BUD31 was commonly overexpressed in SOC and that a high level of BUD31 was associated with poor prognosis, and pan-cancer analysis showed that BUD31 was overexpressed in various cancer types. We next investigated the functional role of BUD31 and found that BUD31 promoted the proliferation and survival of ovarian cancer cells and xenograft tumor growth. In breast cancer cells, BUD31 is required for cell migration[37]. These findings suggest that BUD31 has oncogenic potential and is closely related to the unfavorable prognosis of patients with ovarian cancer.

Synthetic lethality screening provides a promising strategy for cancer therapy[38]. PARP inhibitors are the first clinically approved drugs based on their synthetic lethality in BRCA1/2 mutant ovarian cancer[39]. In MYC-activated state, *BUD31* has been identified as a MYC-synthetic lethal gene, and BUD31 is required for spliceosome assembly and catalytic activity. Depletion of BUD31 in MYC-hyperactive cells leads to global intron retention[26]. However, the BUD31-regulated cancer-specific splicing events and its binding motif remain largely unknown. We performed CLIP-seq to map genome-wide BUD31-RNA interactions and found BUD31-binding sites to be highly enriched in exon-intron regions near both the 3′ and 5′ splicing sites. We further revealed that BUD31 inhibition results in extensive exon skipping. We also identified four BUD31-binding RNA motifs using the HOMER algorithm. In the combined analysis of CLIP-seq, RIP-seq and RNA-seq data, we identified multiple potential direct binding targets (BCL2L12, RBCK1, E2F4, and CDK16) that may involve in BUD31-mediated ovarian cancer cell survival and proliferation. These findings reveal that BUD31 globally regulates AS through direct binding its RNA targets.

BCL2L12 has been identified as a therapeutic target in glioblastoma[33]. RNA interference-based spherical nucleic acids targeting *BCL2L12* has been conducted in a phase 0 first-in-human trial in glioblastoma[34]. BCL2L12 was also reported to interact with BCL2 and BCL-XL in yeast two hybrid system[40]. Intestinally, we found that knockdown of BCL2L12 decreased the protein level of BCL2, whereas there is no significant change in BCL2L12 after BCL2 knockdown, suggesting inactivation of BUD31 led to downregulation of BCL2 may occur as a result of the BCL2L12 downregulation. BCL2L12 is upregulated in human glioblastomas and blocks the activation of caspase-3 and caspase-7 to confer resistance to apoptosis[41,42]. BCL2L12 inhibits p53-dependent DNA damage-induced apoptosis through direct interaction with p53[43]. Importantly, we found that BUD31 knockdown promoted exon 3 skipping that compromises the expression of the full-length isoform *BCL2L12*-L. Although the truncated isoform of *BCL2L12*,

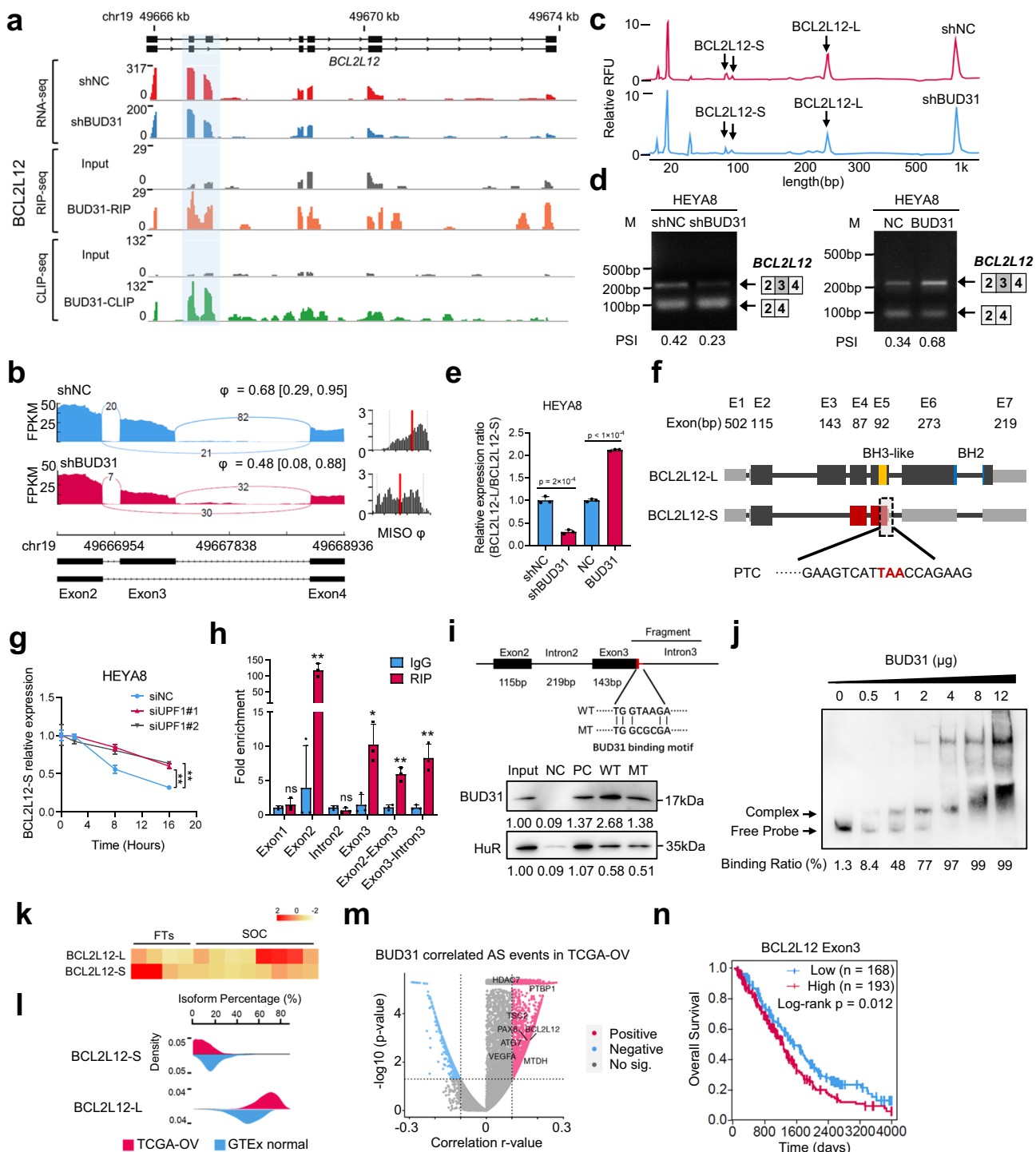

*BCL2L12*-S, is increasingly generated due to BUD31 knockdown, it is rapidly degraded by NMD, and the cells subsequently underwent apoptosis. Mechanistically, BUD31 was shown to bind to exon 3 of *BCL2L12* according to the CLIP-seq and RIP-seq data. BUD31 protein and *BCL2L12* pre-mRNA interactions were verified by EMSA and RNA pull-down assays. More importantly, *BCL2L12*-L was significantly increased in SOCs and higher levels were correlated with poor overall survival. Thus, our work supports the notion that BUD31 stimulates the inclusion of exon 3 to generate *BCL2L12*-L and promotes ovarian cancer progression.

Altered transcriptional regulation might increase BUD31 expression. MYC was shown to regulate the expression of multiple splicing factors including SRSF1[44]. However, we failed to verify BUD31 is a direct

target of MYC in ovarian cancer. We analyzed promoter region of BUD31 and found potential E2F1 binding sites on the BUD31 promoter region, indicating that E2F1 is potential transcriptional factor of BUD31 in ovarian cancer. In addition, posttranscriptional or posttranslational regulation may be also involved in the regulation of BUD31 in ovarian cancer. Nevertheless, future study is needed to determine how BUD31 becomes overexpressed in ovarian cancer.

RNA-based therapeutics is coming of age. SARS-CoV-2 mRNA-based vaccines are approximately 95% effective in preventing COVID-19[45]. Several ASOs have been approved for the treatment of spinal muscular atrophy and Duchenne muscular dystrophy[36]. Spinraza is an FDA-approved drug that can modulate splicing of the *SMN2* gene to generate full-length SMN2 protein and thus improve the motor

**Fig. 6 | BUD31 promotes *BCL2L12* exon 3 inclusion through direct binding to the pre-mRNA. a** The AS pattern and BUD31 direct binding sites in *BCL2L12* were visualized with IGV using RNA-seq, RIP-seq, and SpyCLIP-seq data. The light blue region highlights the alternative exon and the BUD31-binding sites. **b** The Sashimi plots of exon 3 skipping in *BCL2L12* in HEYA8 cells with BUD31 knockdown (red) and corresponding controls (blue). **c, d** Semi-quantitative RT-PCR and fragment analysis were performed to validate AS events in *BCL2L12* after BUD31 knockdown or overexpression. **e** The relative expression ratio of *BCL2L12*-L/*BCL2L12*-S was analyzed in HEYA8 ovarian cancer cells with BUD31 knockdown or overexpression (*n* = 3 biologically independent experiments). **f** Schematic structure of two *BCL2L12* transcripts. *BCL2L12*-L is the full-length transcript, and *BCL2L12*-S is a short transcript lacking exon 3 skipping, which generates a premature termination codon (PTC). **g** *BCL2L12*-S expression was measured by qPCR in UPF1 knockdown and control HEYA8 cells treated with 10 μg/ml actinomycin D at the indicated times. **h** RIP-qPCR was performed to validate the interaction between BUD31 and *BCL2L12*

RNA in HEYA8 cells with the anti-BUD31 antibody (*n* = 3 biologically independent experiments). *$p < 0.05$, **$p < 0.01$. **i** RNA pull-down assay showed the interaction between *BCL2L12* pre-mRNA and BUD31 protein. Protein expression was quantified and normalized with the Input sample. **j** RNA EMSA showed the binding of recombinant BUD31 and *BCL2L12* pre-mRNA fragments. The upper band shows the complex of BUD31 protein and *BCL2L12* pre-mRNA. **k** Relative *BCL2L12*-L/S transcript expression was measured by qPCR in SOC samples (*n* = 8) and FTs (*n* = 4). **l** Isoform percentage of *BCL2L12*-L and *BCL2L12*-S in SOCs and normal ovaries from the TCGA-OV and GTEx datasets. **m** Correlation between *BUD31* mRNA expression and the PSI value of AS events in the TCGA-OV dataset. *P* values and *r* values were calculated by Pearson's correlation. **n** Kaplan–Meier analyzes the correlation between *BCL2L12* exon 3 expression and overall survival based on TCGA data. The *p* values were obtained by two-tailed unpaired Student's *t*-test (**e, g, h**) or log-rank test (**n**), and the results are presented as the mean ± SD. Source data are provided as a Source Data file.

function of spinal muscular atrophy patients[46]. PKM2 is an isoform of the *PKM* gene and plays a crucial role in the Warburg effect in cancer, and splice switching from PKM2 to PKM1 by ASOs restores the sensitivity of cancer cells to chemotherapy[47]. ASO-mediated exon 6 skipping of *MDM4* reduces the amount of full-length MDM4 and inhibits melanoma cell growth[48]. Finally, ASOs have been shown to induce the redirection from Bcl-xL to Bcl-xS and thus to induce apoptosis of melanoma cells in vitro and to inhibit xenograft tumor burden in vivo[20]. Here, we designed ASOs to target *BCL2L12* exon 3 near the 5′ splice sites, which results in exon 3 skipping. We then examined the endogenous *BCL2L12* splicing in A2780 cells transfected with ASOs and found that ASO2 efficiently reduced *BCL2L12*-L levels and increased *BCL2L12*-S levels. Consistently, BCL2L12 protein expression was significantly decreased upon ASO2 treatment. Importantly, ASO2 was able to induce apoptosis and to inhibit the growth of xenograft tumors of ovarian cancer cells both in vitro and in vivo. Our findings thus suggest that ASO-mediated *BCL2L12* exon 3 skipping is a promising strategy for cancer therapy.

In conclusion, our study suggests that BUD31 acts as an oncogenic splicing factor and prognostic marker in ovarian cancer. We further identify the binding motif and the genome-wide binding pattern of BUD31. BUD31 overexpression drives an oncogenic splicing switch in *BCL2L12* to produce full-length *BCL2L12* that in turn confers cancer cells resistance to apoptosis promotes the proliferation of ovarian cancer cells. Inhibition of BUD31 or the use of ASOs may provide potential therapeutic strategies for ovarian cancer.

## Methods

### Nude mouse xenograft model
In the subcutaneous xenograft model, female BALB/c-nude mice (6–8 weeks old) (Vital River Laboratory Animal Technology) were randomly divided into two groups (4–8 mice per group) and injected subcutaneously with doxycycline-induced BUD31 knockdown HEYA8 cells or BUD31 overexpression ID8 cells. Less than 6 mice were housed in a cage at 20–25 °C and 50% humidity with a 12 h light/dark cycle. Doxycycline (1.2 g/L) mixed with 5% sucrose was fed to the experimental group, while 5% sucrose alone was fed to the control group. ASO (Tsingke) intratumor injection was applied after the subcutaneous tumor reached 5 mm in diameter. 5 nmol ASO mixed with 3 μl lipo2000 in 25 μl Opti-MEM was administered every 3 days. When the study finished, the mice were anesthetized, and the tumor volume and weight were measured. The maximal tumor diameter permitted by Shandong University Animal Ethics Research Board is 15 mm and was not exceeded in the experiments.

In the living image xenograft model, female BALB/c-nude mice (6–8 weeks old) were randomly divided into two groups (6 mice per group) and injected intraperitoneally with luciferase-expressing ovarian cancer HEYA8 cells with a dox-inducible BUD31 knockdown system. Administration of doxycycline (1.2 g/L) in the drinking water

started one week after the cell implantation. Three weeks later, the mice were anesthetized with 4% sterile chloral hydrate (7–10 μl/g body weight, Sangon) and D-Luciferin sodium salt (15 mg/ml, dissolved in DPBS, 150 μg/g body weight, Yeason) was injected intraperitoneally. Bioluminescence images were captured 10 min later using an imaging system (PerkinElmer). The Shandong University Animal Ethics Research Board approved the animal experiment procedures (SDULCLL2019-2-08).

### Human tissue samples
SOC specimens and FTs were collected from April 2009 to July 2015 in Qilu Hospital. The SOC samples were obtained from patients with primary ovarian cancer who had not undergone any previous surgery or chemotherapy. In addition, FTs were obtained from patients who underwent total hysterectomy and bilateral salpingo-oophorectomy for uterine diseases or benign neoplastic adnexal pathological changes. Fresh tissue samples were collected within 2 h of surgery and were sliced to 5 mm³ and immersed in 10 vol of RNALater (Ambion, Austin, TX). The tissue samples were stored at –80 °C. All patients provided informed consent, and ethical approval was granted by the Ethics Committee of Shandong University (SDULCLL2019-1-09).

### Cell lines
Human ovarian cancer cell lines A2780 and HEYA8 were obtained from the Jian-Jun Wei lab, Northwestern University. OV90 and OVCAR3 cell lines were purchased from the American Type Culture Collection. The mouse ovarian surface epithelial cell line ID8 was purchased from Sigma-Aldrich (SCC145). HEK293T cell line was obtained from the National Collection of Authenticated Cell Cultures (SCSP-502). Cell lines were validated by STR profiling. A2780, HEYA8, ID8, OV90 and HEK293T cells were cultured in Dulbecco's modified Eagle's medium (DMEM) (Gibco, Invitrogen) containing 10% fetal bovine serum (FBS) (Gibco) and 1% penicillin/streptomycin (Macgene). OVCAR3 cells were cultured in RPMI 1640 (Macgene) supplemented with 20% FBS, 1% penicillin/streptomycin, 1% sodium pyruvate (Sigma), 0.3% glucose (Corning), and 10 ng/mL insulin (Sigma). The cells were cultured at 37 °C in a humidified incubator containing 5% $CO_2$.

### Primary culture of ascites-derived ovarian cancer cells
Patient ascites were collected from the Department of Obstetrics and Gynecology, Qilu Hospital, Shandong University, upon the patient's informed consent. Ethical approval was granted by the Ethics Committee of Shandong University (SDULCLL2019-1-09). Ascites-derived ovarian cancer cells OVBWZX were obtained from a 54 years old female patient diagnosed with high-grade serous ovarian cancer by clinical pathology before receiving chemotherapy and surgical therapy. Primary ovarian cancer cells were cultured according to the previous studies[49,50]. Briefly, after receiving freshly isolated fluid in a sterile

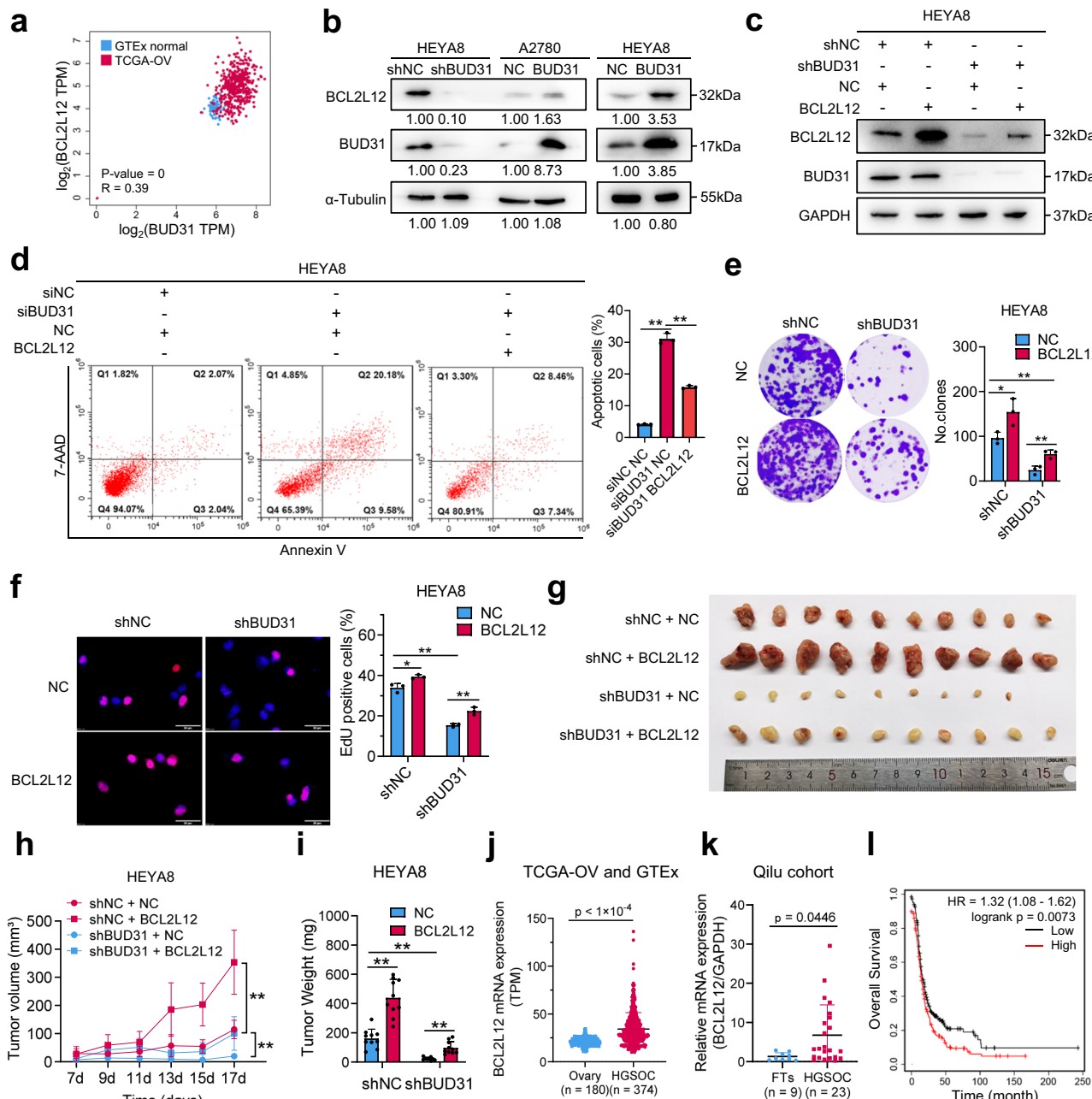

**Fig. 7 | The oncogenic roles of BUD31 in ovarian cancer partially rely on *BCL2L12* expression. a** Correlation analysis of mRNA expression between *BUD31* and *BCL2L12* in SOCs and normal ovaries from the TCGA-OV and GTEx datasets. (Pearson's *r* = 0.39). **b** Western blot analysis of BCL2L12 protein expression in HEYA8 cells with BUD31 knockdown or overexpression in A2780 cells. Protein expression was quantified and normalized with the control group. **c** Immunoblot analysis of BUD31 and BCL2L12 protein levels in HEYA8 cells transfected with shBUD31 or BCL2L12 overexpression vector. Apoptosis (**d**), clonogenic (**e**), and EdU (**f**) assays for investigating the potential of BCL2L12 to rescue the loss of BUD31 in terms of apoptosis and proliferation. **g–i** Xenograft experiments by subcutaneous injection were conducted in HEYA8 cells with dox-inducible BUD31 knockdown or BCL2L12 overexpression vector. Representative image (**g**), volume curves (**h**), and weight (**i**) of xenograft tumors showed that BCL2L12 could partially rescue the inhibitory effect of BUD31 on tumor growth (*n* = 10). **j** BCL2L12 expression was analyzed in SOCs from TCGA-OV (*n* = 374) and in normal ovaries from GTEx (*n* = 180). **k** *BCL2L12* mRNA expression was determined by qPCR in SOC (*n* = 23) and FT (*n* = 9) samples. **l** Kaplan–Meier analysis of BCL2L12 expression on the overall survival of ovarian cancer patients based on cohorts from Kaplan–Meier Plotter. The high and low-expression groups were separated based on the auto-select cutoff. All functional experiments were performed with *n* = 3 biological repeats. The *p* values were determined by a two-tailed unpaired Student's *t*-test (**d**, **e**, **f**, **h**, **i**, **j**, **k**), or log-rank test (**l**), and the results are presented as the mean ± SD. *$p < 0.05$, **$p < 0.01$. Source data are provided as a Source Data file.

vacuum container, 25 ml of ascites fluid was mixed with an equal volume of MCDB/M199 medium in T-75 flasks. Cells were incubated for 3–4 days prior to the first change of complete medium. The medium was then changed every 2–3 days until the flasks were confluent and cells were passaged at a 1:2 dilution. Experiments were performed using cells at passage 2 through passage 6. Primary ovarian cancer cells

were separated into 30 divisions and frozen in 70% v/v MCDB/M199 medium, 20% v/v FBS, and 10% dimethylsulfoxide (DMSO).

**Lentiviral infection and RNA interference**
The BUD31-pENTER overexpression plasmid and the BCL2L12-pENTER plasmid were from WZ Biosciences. The full-length open

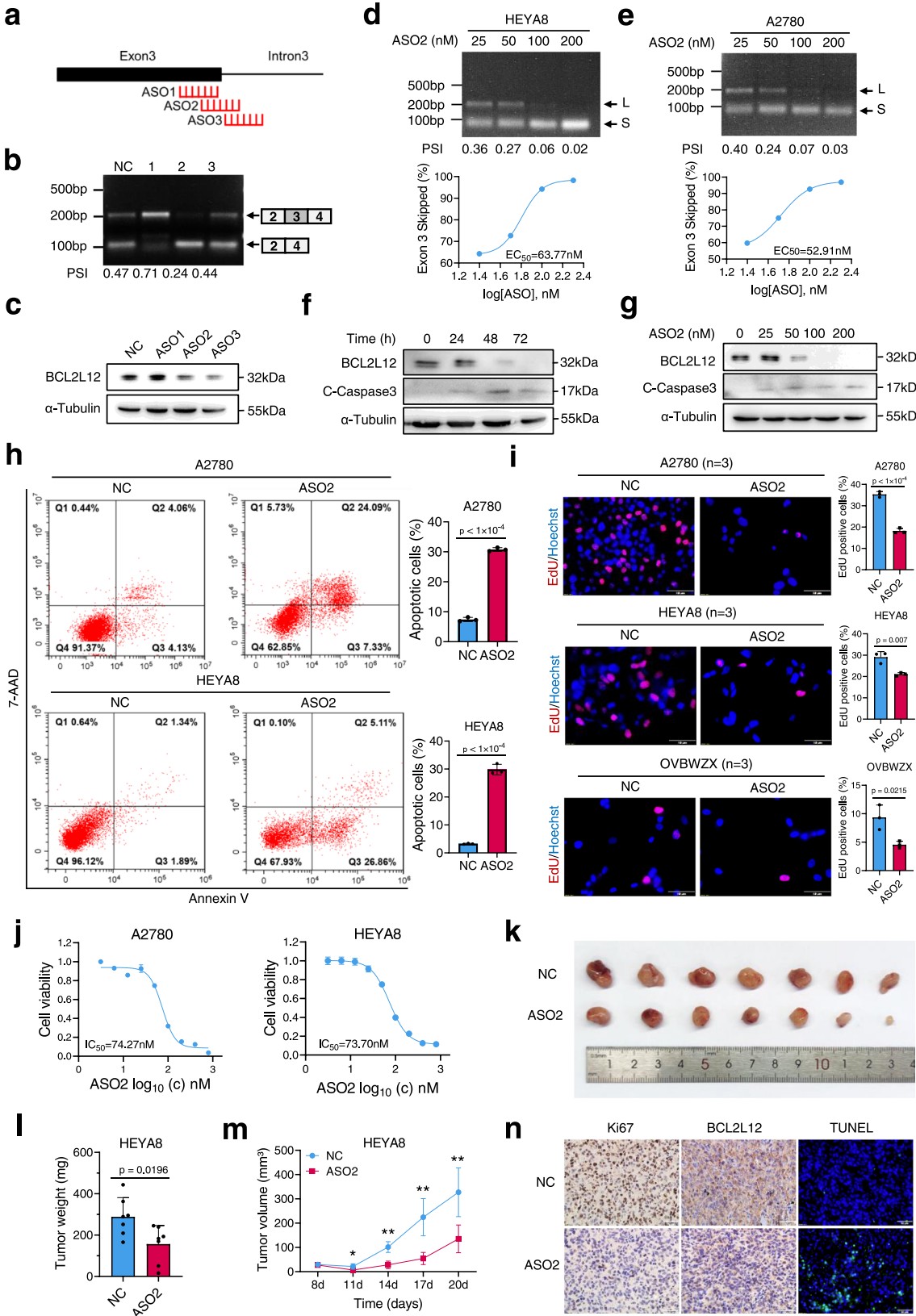

reading frame of *BUD31* was cloned with a ClonExpress®II One Step Cloning Kit (Vazyme) into the modified doxycycline-inducible lentiviral vector pTRIPZ[28]. The shBUD31 sequence targeting BUD31 was cloned into the pZIP-TRE3G plasmid (Transomic). The lentivirus vectors were transfected into HEK293T cells together with psPAX2 and pMD2.G to produce lentivirus particles. Stable cell lines were established by lentivirus infection followed by puromycin (2 μg/ml) selection for 2 weeks. RiboBio prepared a GenOFF st-h-BUD31 suite containing three siRNAs and GenOFF st-h-BCL2L12. Plasmids and siRNAs were transfected into cells using Lipofectamine 2000 reagent

**Fig. 8 | ASO-mediated *BCL2L12* exon skipping induces apoptosis of ovarian cancer cells. a** Schematic diagram of ASO target sites on *BCL2L12* based on the BUD31-binding region on *BCL2L12* as determined from the CLIP-seq data. **b** RT-PCR analysis of the *BCL2L12* AS pattern in response to ASOs. **c** Western blot analysis of the BCL2L12 protein level in A2780 cells transfected with ASOs. Semi-quantitative RT-PCR analysis of the *BCL2L12* AS pattern in HEYA8 (**d**) and A2780 (**e**) cells transfected with ASO2. Dose-dependence curve of ASO2-treated HEYA8 and A2780 cells showing increased skipping of exon 3 [(exon 3 skipped/exon 3 skipped + full-length) × 100] in relation to the log of the dose. The $EC_{50}$ was calculated in HEYA8 (63.77 nM) and A2780 (52.91 nM) cells. BCL2L12 and cleaved-caspase-3 were measured by western blot in A2780 cells treated with 200 nM ASO2 for different times (0, 24, 48, and 72 h) (**f**) and with different concentrations (0, 25, 50, 100, 200 nM) of ASO2 for 72 h (**g**). **h** Apoptotic cells were detected by flow cytometry after staining with Annexin V/7-AAD in A2780 and HEYA8 cells treated with ASO2 (200 nM). **i, j** EdU assay was performed in ovarian cancer cell lines treated with 200 nM ASO2. The $IC_{50}$ was calculated based on the MTT assay. **k–n** ASO2 intratumoral injection to subcutaneous tumor xenografts using HEYA8 cells. The xenograft model showed the inhibitory effect of ASO2 on tumor growth ($n = 6$ mice per group) (**k**). The tumor weight (**l**) and volume (**m**) were measured for each group. The BCL2L12 and Ki-67 expression levels were evaluated with immunohistochemical staining, and the apoptosis level was determined by a TUNEL assay (**n**). All functional experiments were conducted with $n = 3$ biological repeats. The $p$ values were obtained by two-tailed unpaired Student's $t$ test (**h, i, l, m**), and the results are presented as the mean ± SD. *$p < 0.05$, **$p < 0.01$. Source data are provided as a Source Data file.

(Thermo Fisher Scientific) following the manufacturer's instructions. RNA and protein were extracted 48 h or 72 h after transfection or doxycycline induction.

## Recombinant protein expression and purification

*Escherichia coli* BL21(DE3) competent cells (Vazyme) were transformed with SpyCatcher expression plasmid, a gift from Ligang Wu's lab, and the BUD31 expression pET-28a plasmid. A single colony was picked and inoculated into 1 ml Luria-Bertani medium containing 0.5 mg/ml kanamycin for 10 h at 37 °C. A total of 100 µl cells were inoculated and cultured at 37 °C until the absorbance at 600 nm reached 0.4–0.6, and this was followed by induction with 1 mM isopropyl β-D-thiogalactoside (IPTG) at 30 °C for 6 h. Cells were harvested and lysed using an ultrasonic cell crusher. The lysates were centrifuged at $13,400 \times g$ for 50 min, and the soluble fraction was purified with a His-tag Protein Purification Kit according to the manufacturer's protocol. The eluates were dialyzed overnight against dialysis buffer (20 mM Tris-HCl pH 7.8, 150 mM NaCl, 10% glycerol, and 1 mM DTT). SDS-PAGE and Coomassie blue staining checked the purity of each fraction. Purified proteins were aliquoted and stored at −80 °C.

## RNA isolation and PCR analysis

According to the manufacturer's instructions, total RNA was extracted from cultured cells or fresh tissues with a Cell Total RNA Isolation Kit (Foregene). Total RNA was reverse transcribed into cDNAs using HiScript II Q RT SuperMix for qPCR (Vazyme), and qPCR was performed with SYBR Green mix (Vazyme) and the Applied Biosystems QuantStudio 3. GAPDH served as the endogenous control. The $2^{-\Delta\Delta CT}$ method was used for the relative quantification of the qPCR data. Semi-quantitative RT-PCR and Qsep100 Bio-Fragment Analyzer (BiOptic) were used to analyze alternative spliced products. Primer sequences were designed for the constitutively expressed flanking exons[51], and 2 × Taq Master Mix (Dye Plus) (Vazyme, P112-01) was used to simultaneously amplify isoforms that included or skipped the target exon. Primer sequences are listed in Supplementary Table 2.

## Western blot

The samples were lysed on ice in Western and IP Cell Lysis Buffer, and the protein concentration was determined using the bicinchoninic acid protein assay (Beyotime). SDS-PAGE was used to separate protein samples. The membrane was blocked with 5% skim milk before overnight incubation with primary antibodies at 4 °C. All primary antibodies were diluted at a ratio of 1:1000 except anti-alpha tubulin which was diluted at 1:5000. Horseradish peroxidase-conjugated secondary antibodies (1:10,000 dilution, Jackson ImmunoResearch) and an electrochemiluminescence system (GE Healthcare, UK) were used to detect specific proteins. Western blot results were quantified with Fiji (version 1.53c). All antibody information is listed in Supplementary Table 3.

## IP-MS

HEYA8 cells with endogenously expressed BUD31 were harvested, and cell pellets were lysed on ice with Western and IP Cell Lysis Buffer

(Beyotime Institute of Biotechnology). Whole-cell extracts were incubated with 5 µg BUD31 antibody (Proteintech) overnight for 1 h at 4 °C, then incubated with magnetic Protein A/G beads (Millipore) for 2 h at 4 °C. Beads were washed with Western and IP Cell Lysis Buffer three times, and the immunocomplex was resuspended in 1× SDS-PAGE loading buffer and separated followed by Coomassie brilliant blue staining. LC-MS/MS was conducted by PTM-Biolab using a Thermo Fisher LTQ Obitrap ETD. The peptides were confirmed by western blot. The data of the IP-MS are given in Supplementary Data 2.

## Cell proliferation assays

Cell viability and proliferation were determined with a methylthiazolyl diphenyl-tetrazolium bromide (MTT) assay. Cells were seeded in 96-well plates at densities of $1 \times 10^3$ cells per well. After culturing for the designated time, 10 µl of MTT (5 mg/ml) was added to each well and incubated for 4 h at 37 °C. Cell growth was monitored over the following 5 days, and the IC50 was determined 48 h after treatment. The supernatant was discarded after centrifugation, and 100 µl of dimethylsulfoxide was added to each well and the absorbance was measured at 570 nm using a microplate reader (Bio-Rad, Hercules, CA, USA). According to the manufacturer's instructions, an EdU cell proliferation assay was performed using a Cell-Light EdU Apollo567 In Vitro Kit. Briefly, cells were seeded on glass coverslips in 24-well plates at densities of $3–4 \times 10^4$ cells per well and then incubated with the cell culture medium containing EdU for 20–30 min. The cells were then fixed and stained with Apollo567 fluorescent dye and Hoechst 33342.

## Clonogenic assay

Cells were seeded in a six-well plate (1000–2000 cells per well) and cultured for 1–2 weeks. Colonies were fixed with methanol and stained with 0.1% crystal violet (Solarbio), and the number of colonies was counted with Image J. The data are presented as the mean ± SD of three independent experiments.

## Flow cytometry and TUNEL assays

Flow cytometry analysis for cell apoptosis was performed using an Annexin V-PE/7-AAD Apoptosis Detection Kit (Vazyme, A213-01). In $H_2O_2$-induced apoptosis experiments, cells were treated with $H_2O_2$ with a final concentration of 400 µM for 4 h before apoptosis detection. Cells were digested with ethylenediaminetetraacetic acid (EDTA)-free trypsin (Macgene, CC035) for 3 min, collected by centrifugation, washed with ice-cold phosphate-buffered saline (PBS), and resuspended at a density of $5 \times 10^5$ cells/ml with 100 µl 1× binding buffer. Then 5 µl Annexin V-PE and 5 µl 7-AAD were added and incubated for 10 min in the dark. Finally, cells were incubated with an additional 400 µl 1× binding buffer and analyzed within 20 min by CytoFLEX S (Beckman Coulter Life Science). At least $1 \times 10^4$ cells were collected to determine the percentage of apoptotic cells. CytExpert (version 2.4) was used to analyze flow cytometry data. For TUNEL assays, the TUNEL Cell Apoptosis Detection Kit (KEYGEN, KGA703) was used according to the manufacturer's protocol.

## IHC staining of tissue sections

Formalin-fixed and paraffin-embedded tissues or tissue microarray sections were deparaffinized in xylene and rehydrated with a graded series of ethanol solutions. Antigen retrieval was performed in EDTA by heating in a microwave. Tissue slides were blocked with 1.5% normal goat serum and incubated with primary antibodies against Ki-67 (1:200 dilution, CST, 9129S) and BUD31 (1:250 dilution, Proteintech, 11798-1-AP) overnight at 4 °C. The sections were then incubated with the secondary antibody and stained with diaminobenzidine (DAB). The final IHC staining score for the tissue microarray was calculated. Specifically, high (Score ≥ 7) and low (Score < 7) expression of each sample was determined by two pathologists based on the intensity and extent of staining across the tissue microarray section.

## Immunofluorescence staining

HEYA8 and OVBWZX cells were fixed with 4% paraformaldehyde for 30 min at room temperature, followed by permeabilization with 0.5% Triton X-100 in PBS for 15 min at room temperature. Tissue slides were processed as described above for immunohistochemistry staining. Samples were then blocked with 1% BSA for 1 h at room temperature and incubated with primary antibodies against SC35 (1:500 dilution, Abcam, ab204916), BUD31 (1:250 dilution, Proteintech, 11798-1-AP), and α-Tubulin (1:400 dilution, Proteintech, 66031-1-Ig) overnight at 4 °C and then incubated with secondary antibody for 1 h at room temperature in the dark. The images were captured on an Andor Revolution confocal microscope system or an Olympus BX53 microscope system.

## RNA immunoprecipitation (RIP) assay

HEYA8 cells were collected for the RIP assay, which was performed using the EZ-Nuclear RIP Kit (17-701, Merck Millipore) following the manufacturer's instructions. In brief, cells were collected and lysed in RIP lysis buffer, and the lysates were incubated with magnetic beads coated with anti-BUD31 antibody (Proteintech, 11798-1-AP) at 4 °C overnight. The beads combined with immunocomplexes were washed with RIP wash buffer six times and digested by protease K, and RNA was extracted with phenol/chloroform/isoamyl alcohol (125:24:1 mixture). Both input and RIP samples were prepared for next-generation sequencing by Ribobio Biotechnology Company.

## SpyCLIP assay

SpyCLIP was performed according to a previous study[28] with several modifications. Briefly, HEYA8 cell lines transfected with doxycycline-inducible SpyTag-FLAG-BUD31 expression lentivirus were induced with doxycycline for 72 h before harvesting. Cells were crosslinked and irradiated at 400 mJ/cm² in a UV Crosslinker (UVP, CL-1000). Cell nuclei were isolated with a Nucleoprotein Extraction Kit (Sangon) before lysis. Turbo DNase (2 U/μl, Invitrogen, AM2238) and 1:200 diluted RNase I (100 U/μl, Invitrogen, AM2295) were used to remove the DNA and to fragment the RNA. The mixed lysate was incubated with anti-FLAG magnetic beads (MBL, M185-11) at 25 °C for 40 min. After removing RNA-binding proteins from the FLAG beads with phosphoserine phosphatase, the mixture was incubated with fresh pre-washed SpyCatcher beads at 25 °C for 1 h. Stringent washes were conducted according to the manufacturer's instructions, and the beads were digested with proteinase K (Roche, 3115828001). RNA was purified and concentrated with the Spin Column RNA Cleanup & Concentration Kit (Sangon), and the sequencing library was constructed with the NEBNext Ultra RNA Library Prep Kit for Illumina. High-throughput sequencing of the SpyCLIP libraries was performed on a HiSeq 2500 using the PE150 sequencing strategy (Novogene).

## RNA pull-down assay

BCL2L12 pre-mRNA fragments were cloned from human placenta genomic DNA. Primers used for generating wildtype and mutant BCL2L12 fragments and the target sequence are listed in Supplementary Table 2. With the constructed T7 promoter ahead of the target sequence, the RNAMAX-T7 in vitro transcription kit (RiboBio) was used to transcribe the RNA of the BCL2L12 fragment. The fragment was labeled with biotin using the Pierce™ RNA 3′ End Desthiobiotinylation Kit (Thermo Fisher Scientific), and RNA pull-down was performed using the Magnetic RNA Protein Pull-Down Kit (Thermo Fisher Scientific) according to the manufacturer's protocol. The negative control was poly(A)$_{25}$ RNA, and the positive control was the 3′ untranslated region of the androgen receptor RNA. The proteins were detected by western blot analysis.

## RNA EMSA

The RNA EMSA was performed with a CoolShift-BTr RNA EMSA Kit (Viagene) according to the manufacturer's instructions with modifications. Briefly, 10 ng biotin-labeled RNA probe per sample was first heated at 85 °C for 3 min to relax secondary structures. The probe was then incubated with recombinant His-tagged BUD31 at different concentrations for 40 min, and 5% nondenaturing polyacrylamide gel electrophoresis was conducted in 0.25× cool Tris-borate-EDTA (TBE) buffer at 120 V for 70 min. RNA-protein complexes were transferred to the membrane in 0.5× TBE at 390 mA for 40 min. After immobilization and crosslinking with 600 mJ UV in a CL-1000 UV linker, the membrane was blocked and conjugated with HRP. The chemiluminescence was detected with an ECL system (GE Healthcare, Little Chalfont, Buckinghamshire, UK). The binding ratio in RNA EMSA experiment result equals to the complex band intensity compared to the sum value of two bands.

## RNA-seq and quantification

Total RNA was isolated from BUD31 knockdown and control HEYA8 cells (three biological replications of each sample) using TRIzol reagent (Invitrogen) according to the manufacturer's protocol. Poly(A) sequencing libraries were prepared using the Illumina TruSeq-stranded-mRNA protocol after RNA quality was assessed using an Agilent Technologies 2100 Bioanalyzer with the application of an RNA integrity number > 7.0. Adenylated mRNAs were isolated using oligo-d(T) magnetic beads (two rounds). The RNA-seq library was paired-end sequenced using Illumina HiSeq 4000 at LC-Bio. After obtaining paired-end reads, clean reads were aligned to the hg38 genome with HISAT2 (version 2.2.0) and sorted with samtools (version 1.9). Mapped reads were visualized with the Integrative Genomics Viewer (IGV). Transcripts were reconstructed with StringTie (version 1.3.0), and differential expression was analyzed with edgeR (version 3.36.0). The cutoff was set as $q < 0.05$ and fold change (FC) > 1.7 or < 0.6.

## AS and isoform switch analysis

The mapped reads aligned by HISAT2 were used for further analysis, and AS events were identified mainly by rMATS (version 4.1.0)[52]. HEYA8 cells with BUD31 knockdown and corresponding controls ($n = 3$ biological repeats) were used in the analysis. AS events were classified into skipped exon, retained intron, alternative 5′ splice site, alternative 3′ splice site, and mutually exclusive exons. Significant events were filtered out with $p < 0.05$ and |IncLevelDifference| > 0.1. AS events in each sample were also identified with ASprofile[53]. MISO (Mixture of Isoforms, version 0.5.4)[54] analysis was further conducted to confirm the results. Considering the index version of MISO, clean reads were remapped to human genome hg19 with HISAT2. The MISO parameters were --read-len 141 --paired-end 240 117, and BCL2L12 exon 3 skipping was plotted with the sashimi plot program. Global AS analysis was performed as part of the IsoformSwitchAnalyzeR results.

For isoform switch analysis, transcript expression was determined with Salmon (version 0.6.0)[55] using the quant function with -l U -p 8. The quantification results were imported into R for further analysis.

Global isoform fractions and AS events were analyzed with Iso-formSwitchAnalyzeR (version 3.13)[56], an R package stored in Bioconductor. In addition, the coding potentials of the transcripts were determined with the Coding Potential Assessment Tool (version 1.2.1)[57]. Domain information was annotated with Pfam (version 34.0)[58], and Signalp (version 5.0b)[59] was used for signal peptide analysis.

## Coding sequence and UTR length analysis

The elements length of each transcript was determined with SpliceR (version 1.14.0)[60] using the transcripts reconstructed with Cufflink. The enrichment of each transcript was calculated by Cufflink. Length distribution was visualized with a density plot, and statistical differences were calculated inside the R package sm (version 2.2). Cumulative distribution was analyzed by the Kolmogorov–Smirnov test and plotted by ggplot2. NMD sensitivity was determined with SpliceR. For a transcript to be marked as NMD-sensitive, the minimum distance from a stop codon to the final exon-exon junction was 50 nucleotides.

## Identification of SpyCLIP crosslinking sites and binding motif analysis

Crosslinking sites were identified using the previously described iCLIP analysis pipeline[61]. The adapter sequence and low-quality reads were removed with TrimGalore (version 0.6.1), and the quality of the clean reads was checked with FastQC (version 0.11.9). rRNA sequences were removed with bowtie (version 1.2.3)[62]. The remaining reads were mapped to the human genome (hg38) using the STAR software (version 2.7.1a)[63]. PCR duplicates of uniquely mapped reads were removed using Picard (version 2.25.5) with MarkDuplicates. The remaining reads were considered usable reads for identifying crosslinking sites. Mapped reads were visualized in IGV[64]. Two technical replicates of the SpyCLIP samples were merged for PURECLIP (version 1.2.0)[30] analysis with -ld -nt 16 -dm 80. Individual crosslink sites (< 80 nt) were merged as raw binding regions. Binding regions that acquired more than three crosslink sites were used for further analysis as previously suggested[61], and these regions were annotated using HOMER (version 4.11)[65] into exon, intron, promoter, intergenic, 5′UTR, 3′UTR, etc. The 80-nt regions around the center of each binding region were extracted and used to identify the de novo BUD31-binding motif using HOMER's findMotifs program (-len 6,8,10,12 -S 10 -rna -p 4). Motifs were matched to the genome position with scanMotifGenomeWide.pl and function belonging to HOMER visualized with Deeptools (version 3.1.3).

## Visualization of the binding region distributions around the regulated exons

The RNA-seq data upon knockdown of BUD31 in HEYA8 cells were obtained as described above. Alternative splicing events were identified by rMATS (version 4.1.0), and 2000 randomly chosen exons were used as controls. The enrichment of the BUD31-binding signal near the regulated exons was analyzed by deeptools (version 3.1.3) and was calculated using the code downloaded from https://github.com/ulelab/clip-data-science[66].

## Functional enrichment, and Kaplan–Meier survival analysis

The gene expression profiles were obtained from the TCGA and GTEx databases. Differential expression was calculated with reads counts by DESeq2 after normalization. The analysis of the functional enrichment of differentially expressed genes was conducted using PANTHER, GO, and GSEA. GOplot, ggplot2 in R/Bioconductor 3.6.3, and GraphPad Prism 8 were used for plotting. An online Kaplan–Meier plotter database (http://kmplot.com/analysis/) was used to analyze the association between the mRNA expression levels of genes of interest and the survival information of patients with serous ovarian cancer[67]. Cohorts of patients were split by auto-select cutoff. Survival analysis and Kaplan–Meier survival curves were performed in the R packages survival and survminer. High and low-expression groups of our tissue microarray were defined by IHC staining score. All patients with overall survival or progression-free survival information were included.

## Statistics and reproducibility

GraphPad Prism 8 and R (version 3.6.3) were used for statistical analysis. Student's $t$-test and one-way analysis of variance were used to determine significant differences. Pearson's correlation coefficient was used to determine the correlations between gene expression. The chi-square test was used to analyze the differences in clinical characteristics, and the log-rank test was used to detect differences in clinical prognosis. Cumulative distribution was analyzed by the Kolmogorov–Smirnov test. The results are presented as the means ± SD of three independent experiments. Statistical significance was considered as $p < 0.05$. The number of replicates was provided in the figures and legends. Images of EdU, flow cytometry, immunoblots, RNA pulldown, RNA EMSA, IHC, and semi-quantitative RT-PCR were representative of at least two independent experiments.

## Reporting summary

Further information on research design is available in the Nature Research Reporting Summary linked to this article.

## Data availability

The gene expression profiles were obtained from the TCGA (https://portal.gdc.cancer.gov/) and GTEx (https://www.gtexportal.org/) databases. The protein expression profiles were obtained from the CPTAC database (https://proteomics.cancer.gov/programs/cptac). Exon expression and isoform percentage were viewed and downloaded from UCSC Xena (https://xenabrowser.net/). PSI values of AS events in ovarian cancer were collected from TCGASpliceseq[68]. An online Kaplan–Meier plotter database (http://kmplot.com/analysis/) was used in the prognostic analysis. The RNA-seq, RIP-seq, and SpyCLIP data generated in this study have been deposited in the Gene Expression Omnibus (GEO) database under accession code GSE183449, GSE183450, and GSE183451. The IP-MS data generated in this study are provided in Supplementary Data 2. All data are available in the article, Supplementary Information, and source data. Source data are provided with this paper.

## Code availability

The custom code used to analyze the data has been deposited at https://github.com/PrinceWang2018/BUD31_BCL2L12[69]. The analysis pipelines are publicly available as of the date of publication.

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

## Acknowledgements

We thank Translational Medicine Core Facility of Shandong University for consultation and instrument availability that supported this work. We also thank the Laboratory Animal Center of Shandong University for mouse housing and care. This work was supported by the National Natural Science Foundation of China (81972437, 81672578, 82071854).

## Author contributions

Conception and design: Z.W.; Z.L.; C.S.; B.K.. Development and methodology: Z.W.; S.W.; J.Q.; X.Z.; H.G.; V.S.; L.W. Acquisition of data: Z.W.; J.Q.; S.W.; Z.L. Analysis and interpretation of data: Z.W.; X.Z.; Z.L.; C.S.; B.K.; G.L.; H.L.; L.W. Administrative, technical, or material support: C.S.; H.G.; Z.L.; G.L.; H.L.; V.S. Study supervision: Z.L.; C.S.; B.K.; L.W. Writing, review, and/or revision of the manuscript: All authors. Final approval: All authors.

## Competing interests

The authors declare no competing interests.

## Additional information

**Correspondence and requests** for materials should be addressed to Changshun Shao, Beihua Kong or Zhaojian Liu.

