## [Peer Review File · Nature Communications]

Splicing factor BUD31 promotes ovarian cancer progression through sustaining the expression of anti-apoptotic *BCL2L12*Editorial Note: Parts of this Peer Review File have been redacted as indicated to remove third-party material where no permission to publish could be obtained, and to maintain the confidentiality of unpublished data.

REVIEWER COMMENTS

Reviewer #1 (Remarks to the Author):

The manuscript entitled "Splicing factor BUD31 promotes ovarian cancer progression through sustaining the expression of anti-apoptotic BCL2L12" uncovered Bud31 as a potential ovarian cancer gene by regulating cryptic splicing, in particular a gene involved in apoptosis, BCL2L12. Overall the experiments and methods used in the manuscript were comprehensive and generally solid. However, there are also some fundamental challenges to validate the key findings of the manuscript, e.g. the selective essentiality of BUD31 in ovarian cancer, and the critical target genes of BUD31 in addition to BCL2L12. Further clarification of these points is crucial to justify the impact of the key findings in the manuscript.

Specific points:

1. Figures 1A,B,C: the subunits of the highly conserved SF3b subcomplex, SF3B1 and SF3B4, demonstrated opposite differential trend in all analyses. Do authors have an explanation on it? These data will certainly challenge the interpretation and meaningfulness of the role of these genes in all of these parameters (differential expression, OS and PFS).

2. In the DepMap CRISPR and RNAi screenings in 700+ cancer cell lines, BUD31 was clearly shown as a pan-essential gene in all cell lines and lineages (see below). And ovarian cancer cell lines are amongst the least sensitive lineage, although still essential. This challenges the significance of the key findings in the manuscript: is BUD31 really a ovarian cancer (selective) gene, or simply a commonly essential gene that will provides similar roles (dependency) as proposed in some studies in the manuscript?

3. Figures 5,6: There are thousands of genes/junctions that are regulated by BUD31 in the study. While the authors have shown quite convincing data of the regulation and functional relevance of BCL2L12, I wonder if there are other genes/junctions are of great importance. For instance, PAX8, an essential lineage TF in ovary/ovarian cancer, appears to be correlated to BUD31 as well (Fig. 6M). Can authors further clarify if this (or any other) target gene contributes to BUD31-related cancer-promoting activities?

Reviewer #2 (Remarks to the Author):

The authors nicely presented various data demonstrating that the splicing factor BUD31 enhances ovarian cancer progression via including exon3 of BCL2L12, followed by stable expression of the oncogenic protein. The analyses were extensively performed, including TCGA, RNA-seq, CLIP-seq, and splicing isoforms. Moreover, the data were well validated by KD, OE, and ASO treatment. We believe the results shown in this study provide valuable evidence and stimulate further validation in the future. However, there are still several concerns to be addressed.

Major points.

Considering that BUD31 is highly expressed and such upregulation is associated with poor prognosis, the most reasonable strategy should focus on how the OE of BUD31 alters the splicing and consequent cellular phenotypes. However, the authors mainly stick to the KD system. I understand that this is the right way to evaluate the role of endogenous BUD31, but the authors should address how upregulated BUD31 affects the splicing of entire pre-mRNA (Especially in Figure 5A).

In Figures 3A and 3B, the authors should show both BUD KD and BUD31 OE data of HEYA8 to better data integrity.

In Figure 2D, the authors should show BUD31 OE data of HEYA8 cells to clear BUD31 KD or OE effects in the syngenic background.

Figure 2 shows that the knockdown of BUD31 induces apoptosis. The authors should show the

validation of knockdown of BUD31 at E and D, at least 2-3 different siRNA/shRNA data or addback experiment of siRNA/shRNA-resistant BUD31 cDNA. Also, the authors should show the sequence of siRNA.

As spliceosomal proteins are so abundant and IP/MS tends to result in false-positive interactomes, the validation of FigS3 should be carefully interpreted. For example, it looks like SF3B1-BUD31 is validated. However, SF3B1 is one of the most significantly downregulated genes in TCGA-OV (Fig1). The direct interaction on pre-mRNA would be hard to be certified in the current experiments. Please describe the concept of spliceosome disruption in ovarian cancer from BUD31 data.

In the splicing/NMD study, the volume of mature mRNA and/or proteins is particularly important. In Figure 5F, the author should perform any GO/pathway analysis to see if NMD targets have specific roles in the phenotype.

About BCL2L11 splicing, what happens if BUD31 is overexpressed. In Figure 6E, the effects on the L/S ratio look modest. Carefully evaluate the protein expression since skipping exon3 causes NMD, not gain-of-function protein. If BUD31-OE promoted the BCL2L11 protein level (Fig7B), what is the mechanism for that? Is this depending on the extent of exon3 inclusion?

Minor points.

In Figure 4C, the quantification of co-localization would be necessary.

I would suggest the author use the same color for the control group (Fig3B).

In Figure 5A, please show the data from another cell line to confirm the pan-ovarian effect of BUD KD.

In Figure 6G, the authors should validate the siUPF1 data. Please show the data from 2 different siUPF1.

In Figure 7B, please show BUD31 OE results of HEYA8 cells.

In the Discussion, please give some mechanistic ideas on how BUD31 is overexpressed in ovarian cancer?

In Figures 6I and J, better quality data would be required.

Reviewer #3 (Remarks to the Author):

In this manuscript, the authors claimed that the splicing factor BUD31 promotes ovarian cancer progression through sustaining the expression of BCL2L12. There are quite a few questions and inconsistency in this manuscript.

1. In Figure 1 J, K vs. H, I. The survival curves in the Kaplan-Meier Plotter cohort are not as striking as the Qilu cohort. Why?
2. In Figure 2, knockdown of BUD31 caused a significant downregulation of BCL2. What is the mechanism of this? Why did the authors not find the downregulation of BCL2 in later experiment but identified BCL2L12 as the target of BUD31? BCL2 is one of the most important anti-apoptotic proteins in the cells to regulate apoptosis.
3. In Fig. 2, the induced cleaved-caspase 3 and cleaved PARP suggest a much higher level of cell death (Fig. 2E) after knockdown of BUD31, which is not in consistent with the cell death observed in Fig. 2D.
4. In Fig.3, the authors also found that BUD31 promotes cell proliferation. What are the potential mechanisms? What is the relationship between cell proliferation and cancer progression?

5. In Fig 7, the authors found that BCL2L12 overexpressing inhibit cell death (Fig. 7D). How BCL2L12 inhibit cell death? Previous studies have indicated that BCL2L12 as anti-apoptotic protein, can the authors give some possible mechanisms of how BCL2L12 inhibit cell death?
6. The authors claimed that BUD31 promotes ovarian cancer progression through the expression of BCL2L12. This claim can not be verified by just using the overexpression of BCL2L12. A knockdown or a knockout of BCL2L12 should also be performed.
7. What is the relationship between BCL2L12 downregulation vs. BCL2 downregulation in the inhibition of cancer progression?

Response to reviewers' comments

We thank the reviewers for their constructive and thoughtful comments, which have helped us to improve the manuscript substantially. We have addressed reviewer's concerns by performing new experiments, refining analyses of CLIP-seq and RNA-seq data and revising the related text. The following is a point-by-point response to the reviewers' comments.

Response to Reviewer #1 (Remarks to the Author)

----- Reviewer comments:

The manuscript entitled "Splicing factor BUD31 promotes ovarian cancer progression through sustaining the expression of anti-apoptotic BCL2L12" uncovered Bud31 as a potential ovarian cancer gene by regulating cryptic splicing, particularly a gene involved in apoptosis, BCL2L12. Overall the experiments and methods used in the manuscript were comprehensive and generally solid. However, there are also some fundamental challenges to validate the key findings of the manuscript, e.g. the selective essentiality of BUD31 in ovarian cancer, and the critical target genes of BUD31 in addition to BCL2L12. Further clarification of these points is crucial to justify the impact of the key findings in the manuscript.

Response: We thank the reviewer for the thoughtful review of our work and the constructive comments that helped us to improve the manuscript.

To better explain the rationale for finding the selective essentiality of BUD31 in ovarian cancer, we conducted an RNAi screen comprising SF3B1 and five splicing factors (BUD31, SF3B4, SRSF4, MATR3, CHERP) associated with poor prognosis in ovarian cancer patients. Both BUD31 and SF3B1 are found to be essential for ovarian cancer cell survival (**Revised Figure 1D and 1E**). Considering that higher BUD31 expression predicts a worse prognosis, we chose BUD31 for further investigation. We have added this information to the revised manuscript (**Lines 80-83**).

To identify critical target genes of BUD31 in addition to BCL2L12, we optimized bioinformatics analysis on CLIP-seq, RIP-seq, and RNA-seq data and found 317 candidate targets. RBCK1 (another apoptotic-related gene), E2F4, and CDK16 (proliferation-related genes) were validated (**Revised Figure 5I-5R**) as AS targets of BUD31.

Our responses to specific concerns are given below.

Specific points:

1. Figures 1A,B,C: the subunits of the highly conserved SF3b subcomplex, SF3B1 and SF3B4, demonstrated opposite differential trend in all analyses. Do authors have an explanation on it? These data will certainly challenge the interpretation and meaningfulness of the role of these genes in all of these parameters (differential expression, OS and PFS).

Response: We thank the reviewer for the insightful comment. The SF3B complex is an essential component of the U2 snRNP responsible for branch point recognition, spliceosome assembly and activation (Obeng EA, Cancer Cell, 2016) ¹. The SF3B complex contains seven components and SF3B1 is the most commonly mutated splicing factor across cancers (Liu, Z, Cancer Cell, 2020) ².

As the reviewer points out, SF3B1 and SF3B4 exhibited opposite differential trends in all analyses. To answer the question raised by the reviewer, we analyzed the expression pattern of seven family members of the SF3B complex in ovarian cancer or normal tissues using TCGA and GTEx data. We found that the expression level of SF3B1 was decreased, whereas SF3B4, SF3B5, SF3B6, and SF3B7 were increased in ovarian cancer tissues compared to normal tissues. Consistent with our analysis, a previous study shows that SF3B1 expression is decreased during hematopoietic development, while SF3B2, SF3B3, SF3B4, and SF3B5 are upregulated (Li, Y, EMBO Rep, 2021) ³. **We speculate that the loss of SF3B1 may be compensated by other members of the SF3B complex with overlapping functions.**

2. In the DepMap CRISPR and RNAi screenings in 700+ cancer cell lines, BUD31 was clearly shown as a pan-essential gene in all cell lines and lineages (see below). And ovarian cancer cell lines are amongst the least sensitive lineage, although still essential. This challenges the significance of the key findings in the manuscript: is BUD31 really a ovarian cancer (selective) gene, or simply a commonly essential gene that will provides similar roles (dependency) as proposed in some studies in the manuscript?

[REDACTED]

Response: We thank the reviewer for raising this concern and providing an analysis of BUD31 using DepMap. Based on our data (**Revised Figure S1C**) and previous report on breast cancer cells (Koedoot, E, Nat Commun, 2019) ⁴. **We propose that spliceosome component BUD31 is a commonly essential gene.** We also analyzed BUD31 in ovarian cancer cell lines using DepMap. We found BUD31 is essential in most types of ovarian cancer cell lines except for two hyper-mutated cell lines IGROV1 (Domcke S, Nat Commun, 2013) ⁵ and OC314 (Mitsopoulos C, Nucleic Acids Res, 2021) ⁶. In our study, we found that knockdown of BUD31 induced spontaneous apoptosis in ovarian cancer cell lines, indicating the essential role of BUD31 in ovarian cancer.

3. Figures 5,6: There are thousands of genes/junctions that are regulated by BUD31 in the study. While the authors have shown quite convincing data of the regulation and functional relevance of BCL2L12, I wonder if there are other genes/junctions are of great importance. For instance, PAX8, an essential lineage TF in ovary/ovarian cancer, appears to be correlated to BUD31 as well (Fig. 6M). Can authors further clarify if this (or any other) target gene contributes to BUD31-related cancer-promoting activities?

Response: We agree with the reviewer that BUD31 has other important splicing targets besides BCL2L12. We optimized bioinformatics analysis of CLIP-seq, RIP-seq, and RNA-seq data and found 317 candidate targets. RBCK1 (HOIL1), E2F4, and CDK16 were validated (**Revised Figure 5I-5R**). RBCK1 is a component of the linear ubiquitin chain assembly complex (LUBAC), and loss of RBCK1 induces caspase-8 dependent apoptosis (Jain, R, Cell Death Differ, 2021)⁷. Our data revealed that knockdown RBCK1 induced apoptosis of ovarian cancer cells (**Revised Figure 5R**).

As the reviewer points out, BUD31 was correlated with alternative splicing events of PAX8 in TCGA ovarian cancer data (**Original Figure 6M**). PAX8 is an oncogenic transcription factor in high-grade ovarian cancer and splicing variants of PAX8 have been reported previously (Chaves-Moreira, Cancer Res, 2021)⁸. However, PAX8 was not found in BUD31-bound transcripts in our CLIP-seq data, and we failed to validate PAX8 as the direct alternative splicing target of BUD31 in ovarian cancer cells. Other

splicing factors may be involved in the regulation of PAX8 alternative splicing. We now provide more evidence to confirm the specificity of BUD31 splicing on BCL2L12 as compared to other targets. Using high-stringent criteria, we performed integrated analysis of RNA-seq data for differential alternative splicing events using Spliceseq, MISO and rMATS and found 105 AS events. BCL2L12 is one of the top ten alternatively spliced genes, which is also bound with BUD31 (CLIP-seq and RIP-seq). Phenotypically, knockdown of BUD31 induced spontaneous apoptosis, which could be rescued by overexpression of BCL2L12. We have added more data as suggested by the reviewers to prove that BUD31 promotes BCL2L12 exon 3 inclusion through direct binding to the pre-mRNA. Importantly, we performed additional RNA-seq in BUD31 knockdown and control SKOV3 and A2780 cells and still found that BUD31 knockdown promoted exon 3 skipping of BCL2L12. Moreover, we conducted RIP-seq in mouse spermatogenic cells (GSE189715) and BCL2L12 was found to be bound by BUD31. RNA-seq of spermatogenic cells in Bud31 conditional knockout and control mice (GSE189714) revealed that BCL2L12 was significantly decreased upon loss of BUD31. These data suggest that the BCL2L12 is a direct downstream target of BUD31-mediated splicing regulation in ovarian cancer.

Response to Reviewer #2

----- Reviewer comments:

The authors nicely presented various data demonstrating that the splicing factor BUD31 enhances ovarian cancer progression via including exon3 of BCL2L12, followed by stable expression of the oncogenic protein. The analyses were extensively performed, including TCGA, RNA-seq, CLIP-seq, and splicing isoforms. Moreover, the data were well validated by KD, OE, and ASO treatment. We believe the results shown in this study provide valuable evidence and stimulate further validation in the future. However, there are still several concerns to be addressed.

Response: We appreciate the reviewer's recognition of our work and valuable comments that help us improve the manuscript's quality. Our point-to-point responses

are provided below.

Major points.

1. Considering that BUD31 is highly expressed and such upregulation is associated with poor prognosis, the most reasonable strategy should focus on how the OE of BUD31 alters the splicing and consequent cellular phenotypes. However, the authors mainly stick to the KD system. I understand that this is the right way to evaluate the role of endogenous BUD31, but the authors should address how upregulated BUD31 affects the splicing of entire pre-mRNA (Especially in Figure 5A).

Response: We thank the reviewer for this remark. To address the effect of upregulated BUD31 on splicing of entire pre-mRNA, we conducted new RNA-seq in HEYA8 cells with or without BUD31 overexpression and analyzed AS events by rMATS. Interestingly, overexpression and knockdown of BUD31 had opposite effects on AS. Overexpression of BUD31 resulted in less exon skipping (57%) compared to knockdown of BUD31 (68%), suggesting that BUD31 at a high level promotes exon inclusion. We have added these data to the revised figures (**Figure 5B and 5D**) and the revised manuscript (**Lines 185-187**). The RNA-seq data have been deposited in GEO database (GSE183449).

2. In Figures 3A and 3B, the authors should show both BUD KD and BUD31 OE data of HEYA8 to better data integrity.

Response: According to the reviewer's suggestion, we have now provided a new set of data regarding HEYA8 cells with BUD31 knockdown and overexpression (**Revised Figures 3A and 3B**).

3. In Figure 2D, the authors should show BUD31 OE data of HEYA8 cells to clear BUD31 KD or OE effects in the syngenic background.

Response: We have added data of HEYA8 cells as suggested in the revised manuscript (**Revised Figure 2E**).

4. Figure 2 shows that the knockdown of BUD31 induces apoptosis. The authors should show the validation of knockdown of BUD31 at E and D, at least 2-3 different siRNA/shRNA data or addback experiment of siRNA/shRNA-resistant BUD31 cDNA. Also, the authors should show the sequence of siRNA.

Response: Thanks for this suggestion. We have conducted these experiments using two siRNAs against BUD31 in HEYA8 and OV90 cells (**Revised Figures 2D and 2F**). The knockdown efficiency was determined by immunoblotting (**Revised Figure S2F**). Sequences of siRNA used in this study are shown in **Table S5**.

5. As spliceosomal proteins are so abundant and IP/MS tends to result in false-positive interactomes, the validation of FigS3 should be carefully interpreted. For example, it looks like SF3B1-BUD31 is validated. However, SF3B1 is one of the most significantly downregulated genes in TCGA-OV (Fig1). The direct interaction on pre-mRNA would be hard to be certified in the current experiments. Please describe the concept of spliceosome disruption in ovarian cancer from BUD31 data.

Response: We agree with the reviewer that the interaction between BUD31 and SF3B1 is likely to be false-positive, and we have removed it from the original **Figure S4B**.

We appreciate this important comment and rephrased the concept of spliceosome disruption in ovarian cancer from BUD31 data as follows: Unlike the high mutation rate of spliceosome components in hematological malignancies, ovarian cancer exhibits frequent alterations in components of the spliceosome machinery including BUD31, which lead to alternative splicing changes in cancer specific genes involved in cancer cell survival and cancer progression. Targeting the dysregulated spliceosome is a promising therapeutic strategy.

6. In the splicing/NMD study, the volume of mature mRNA and/or proteins is particularly important. In Figure 5F, the author should perform any GO/pathway analysis to see if NMD targets have specific roles in the phenotype.

Response: We are grateful for the suggestion. We performed pathway enrichment analysis on NMD targets and found that the mitotic cell cycle process, DNA replication, and apoptotic signaling pathways were enriched. We have included the new data (**Figure 5H**) and revised the manuscript accordingly.

7. About BCL2L12 splicing, what happens if BUD31 is overexpressed. In Figure 6E, the effects on the L/S ratio look modest. Carefully evaluate the protein expression since skipping exon3 causes NMD, not gain-of-function protein. If BUD31-OE promoted the BCL2L12 protein level (Fig7B), what is the mechanism for that? Is this depending on the extent of exon3 inclusion?

Response: We thank the reviewer for bringing this point to our attention. The modest effects of BUD31 overexpression on the L/S ratio might be due to endogenous expression of BUD31. To further verify the regulation alternative splicing of BCL2L12 by BUD31, we performed semi-quantitative RT-PCR in HEYA8 and A2780 cells upon BUD31 overexpression. We found that overexpression of BUD31 promoted exon 3 inclusion and led to increased generation of full-length isoform and decreased generation of truncated isoform of BCL2L12 (**Revised Figure 6D-6E and Figure S6A-S6B**). Further immunoblotting revealed that forced expression of BUD31 increased the protein level of BCL2L12 (**Revised Figure 7B**). We have included the new data and revised the manuscript accordingly.

Minor points.

1. In Figure 4C, the quantification of co-localization would be necessary.

Response: Following the advice of the reviewer, co-localization of BUD31 and SC35 immunostaining was quantified (**Revised Figure 4C**).

2. I would suggest the author use the same color for the control group (Fig3B).

Response: This has been changed as suggested (**Revised Figure 3B**).

3. In Figure 5A, please show the data from another cell line to confirm the pan-ovarian effect of BUD KD.

Response: According to the reviewer's suggestion, we performed new RNA-seq on BUD31 knockdown A2780 and SKOV3 cells, and alternative splicing events were

analyzed with the same method and threshold. We found that 1376 (56%) AS genes of HEYA8 cells upon BUD31 knockdown were verified in A2780 or SKOV3 cells, and 463 genes with AS events were detected in all three cell lines (**Revised Figure S5A**). Consistently, the predominant AS event upon BUD31 knockdown was exon skipping, accounting for 61% in A2780 cells and 63% in SKOV3 cells (**Revised Figures S5B and S5C**). Importantly, we still found that BUD31 knockdown promoted exon 3 skipping of *BCL2L12* when we analyzed RNA-seq data in A2780 and SKOV3 cells upon BUD31 knockdown. We have included the new data and revised the manuscript accordingly. The RNA-seq data have been deposited in GEO database (GSE183449).

4. In Figure 6G, the authors should validate the siUPF1 data. Please show the data from 2 different siUPF1.

Response: Thank you for this suggestion. We have conducted these experiments using two siRNAs against UPF1 and added the new data to revised **Figures 6G, S6C, and S6D**.

5. In Figure 7B, please show BUD31 OE results of HEYA8 cells.

Response: According to the reviewer's comment, immunoblotting analyses were performed to examine the protein level of BCL2L12 in HEYA8 cells upon BUD31 overexpression (**Revised Figure 7B**).

6. In the Discussion, please give some mechanistic ideas on how BUD31 is overexpressed in ovarian cancer?

Response: We appreciate this constructive suggestion. We have added a detailed discussion of the possible mechanism of BUD31 overexpression in ovarian cancer as follows:

Increase BUD31 expression might be caused by increased transcription. MYC was shown to regulate the expression of multiple splicing factors, including SRSF1 (Da S, Cell Rep, 2012)⁹. However, we failed to verify that BUD31 is a direct target of MYC in ovarian cancer. We analyzed the promoter region of BUD31 and found potential E2F1 binding sites on the BUD31 promoter region, indicating E2F1 might act as a potential transcription factor of BUD31 in ovarian cancer. Posttranscriptional or posttranslational regulation may also be involved in regulating BUD31 in ovarian cancer. We have added this information to the revised manuscript (**Lines 354-361**).

7. In Figures 6I and J, better quality data would be required.

Response: We have replaced original **Figures 6I and 6J** with better quality data and quantified the bands.

Response to Reviewer #3

----- Reviewer comments:

General comments:

In this manuscript, the authors claimed that the splicing factor BUD31 promotes ovarian cancer progression through sustaining the expression of BCL2L12. There are quite a few questions and inconsistency in this manuscript.

Response: We thank you for reviewing our manuscript and asking the many important

questions. Our point-to-point responses are given below.

Major points:

1. In Figure 1 J, K vs. H, I. The survival curves in the Kaplan-Meier Plotter cohort are not as striking as the Qilu cohort. Why?

Response: Thank you for this insightful comment. The Kaplan–Meier curves of ovarian cancer shown in original **Figures 1J and 1K** were based on RNA expression (RNA-seq or microarray) using the Kaplan–Meier Plotter cohort. However, the Kaplan–Meier curves of ovarian cancer shown in original **Figures 1H and 1I** were based on immunohistochemistry (IHC). We believe that the IHC data reflect the expression levels more directly than the RNA data and may better predict the prognosis.

There are many other factors that might affect the relationship between gene expression levels and survival curves, such as patient cohorts (patients are derived from multiple datasets of Kaplan–Meier Plotter cohort), data entry, and cut-off threshold.

2. In Figure 2, knockdown of BUD31 caused a significant downregulation of BCL2. What is the mechanism of this? Why did the authors not find the downregulation of BCL2 in later experiment but identified BCL2L12 as the target of BUD31? BCL2 is one of the most important anti-apoptotic proteins in the cells to regulate apoptosis.?

Response: We thank the reviewer for this important comment. As the reviewer points out, BCL2 protein level significantly decreased upon BUD31 knockdown. However, no significant decrease in BCL2 mRNA expression was observed in RNA-seq analysis of BUD31 knockdown and control HEYA8 cells. Therefore, we speculate that the BCL2 protein level is down-regulated by ubiquitin-proteasome system mediated degradation in BUD31 knockdown-induced ovarian cancer cell apoptosis based on previous report (Edison N, Cell Rep, 2017) ¹⁰.

We completely agree with the reviewer that BCL2 is one of the most anti-apoptotic proteins, and we do hope BCL2 is a direct target of BUD31. BCL2 has three exons, and two isoforms (BCL-2 α and BCL-2 β) are reported (Warren CFA, Cell Death Dis, 2019)¹¹.

However, BCL2 was not included in BUD31-bound genes and AS-related genes based on CLIP-seq and RNA-seq data analysis. Many studies showed that the downregulation of BCL2 and, correspondingly, the upregulation of BAX mostly represent a pro-apoptotic propensity. It appears that a reduced ratio of BCL2/BAX can occur independent of the upstream events. This reduced ratio reflects the execution as well as the outcome of increased apoptosis.

BCL2L12 has been identified as a therapeutic target in glioblastoma, which impedes p53-dependent DNA damage-induced apoptosis (Stegh, AH, Genes Dev, 2010) ¹². BCL2L12 interacted directly with and neutralized caspase-7 and neutralized caspase-3 by an indirect mechanism (Stegh, AH, PNAS, 2008) ¹³. Recently, RNA interference-based spherical nucleic acids targeting BCL2L12 have been conducted in a phase 0 first-in-human trial in glioblastoma (Kumthekar P, Sci Transl Med, 2021) ¹⁴. These studies indicate that BCL2L12 has important anti-apoptotic roles. Our study here identified BCL2L12 as a functional splicing target of BUD31.

3. In Fig. 2, the induced cleaved-caspase 3 and cleaved PARP suggest a much higher level of cell death (Fig. 2E) after knockdown of BUD31, which is not in consistent with the cell death observed in Fig. 2D.

Response: Thank you for raising this concern. In **original Figure 2D**, apoptosis was measured by Annexin V-PE/7-AAD staining in ovarian cancer cells 72h after knockdown of BUD31, whereas in **original Figure 2E**, apoptosis-related proteins were measured at 96h after knockdown of BUD31. Different time points or methods might cause results inconsistency between these two panels. We have now repeated these experiments and provided a new set of data (**Revised Figures 2D and 2E**).

4. In Fig.3, the authors also found that BUD31 promotes cell proliferation. What are the potential mechanisms? What is the relationship between cell proliferation and cancer progression?

Response: As a splicing factor, BUD31 regulates alternative splicing of multiple targets.

We analyzed RNA-seq data of HEYA8 cells with or without BUD31 overexpression and found that cell proliferation-related pathways were significantly enriched (**Revised Figure 2A**). We reanalyzed CLIP-seq, RIP-seq, and RNA-seq data and verified E2F4 and CDK16 as AS targets of BUD31 (**Revised Figure 5K and 5N**). These data suggest BUD31 promotes cell proliferation through regulating proliferation-related genes, including E2F4 and CDK16.

Acquired capability for sustaining proliferative signaling is one of the hallmarks of cancer (Hanahan D, *Cancer Discov*, 2022) ¹⁵. Proliferation is an essential part of cancer progression involved in tumor growth, recurrence, and cancer stem cell self-renewal. For example, HER2 is a tyrosine kinase and, when activated, promotes cell proliferation. HER2 amplification has been associated with poor prognosis in many malignancies (Krishnamurti U, *Adv Anat Pathol*, 2014) ¹⁶. HER2 is an established therapeutic target in various kinds of cancers (Oh DY, *Nat Rev Clin Oncol*, 2020) ¹⁷.

5. In Fig 7, the authors found that BCL2L12 overexpressing inhibit cell death (Fig. 7D). How BCL2L12 inhibit cell death? Previous studies have indicated that BCL2L12 as anti-apoptotic protein, can the authors give some possible mechanisms of how BCL2L12 inhibit cell death?

Response: As the reviewer points out, our data revealed that BCL2L12 exhibits an anti-apoptotic effect in ovarian cancer cells. BCL2L12 is a member of the anti-apoptotic family, which interacts with BCL2 and BCL-XL (Yang MC, *Int J Oncol*, 2015) ¹⁸. BCL2L12 inhibits p53-dependent DNA damage-induced apoptosis (Stegh, AH, *Genes Dev*, 2010) ¹². BCL2L12 is also shown to be a nuclear and cytoplasmic oncoprotein that blocks the activation of caspase-3 and caspase-7 (Stegh, AH, *PNAS*, 2008) ¹⁹.

6. The authors claimed that BUD31 promotes ovarian cancer progression through the expression of BCL2L12. This claim cannot be verified by just using the overexpression of BCL2L12. A knockdown or a knockout of BCL2L12 should also be performed.

Response: We appreciate this constructive suggestion and have added the BCL2L12

knockdown experiments (**Revised Figures S7A-S7D**).

7. What is the relationship between BCL2L12 downregulation vs. BCL2 downregulation in the inhibition of cancer progression?

Response: As mentioned above, downregulation of BCL2 represent a pro-apoptotic propensity and may occur independent of the upstream events. We speculate that just as BCL2 is downregulated when cancer cells, or many other types of cells, are encountering adversary events such as chemotherapeutic drug and radiation, and thus renders cancer cells more prone to apoptosis, the downregulation of BCL2 may occur as a result of the BCL2L12 downregulation.

As suggested, we used siRNAs to knock down BCL2L12 and BCL2 and investigated their roles in ovarian cancer. We found that knockdown of BCL2L12 and BCL2 induced spontaneous apoptosis, and there is more proportion of apoptotic cells in siBCL2 than in siBCL2L12 cells (**Revised Figure S7I**). Interestingly, knockdown of BCL2L12 significantly decreased the protein level of BCL2, whereas there is no significant change in BCL2L12 after BCL2 knockdown (**Revised Figures S7G and S7H**). These data suggest that BCL2L12 appears to be upstream in the BCL2 apoptotic pathway. We have now included these data in the revised figure and revised manuscript (**Lines 281-283**).

References

1. Obeng EA, *et al.* Physiologic Expression of Sf3b1(K700E) Causes Impaired Erythropoiesis, Aberrant Splicing, and Sensitivity to Therapeutic Spliceosome Modulation. *Cancer Cell* **30**, 404-417 (2016).
2. Liu Z, *et al.* Mutations in the RNA Splicing Factor SF3B1 Promote Tumorigenesis through MYC Stabilization. *Cancer Discov* **10**, 806-821 (2020).
3. Li Y, *et al.* A splicing factor switch controls hematopoietic lineage specification of pluripotent stem cells. *EMBO Rep* **22**, e50535 (2021).
4. Koedoot E, *et al.* Uncovering the signaling landscape controlling breast cancer cell migration identifies novel metastasis driver genes. *Nat Commun* **10**, 2983 (2019).
5. Domcke S, Sinha R, Levine DA, Sander C, Schultz N. Evaluating cell lines as tumour models by comparison of genomic profiles. *Nat Commun* **4**, 2126 (2013).
6. Mitsopoulos C, *et al.* canSAR: update to the cancer translational research and drug discovery knowledgebase. *Nucleic Acids Res* **49**, D1074-D1082 (2021).
7. Jain R, *et al.* Dual roles for LUBAC signaling in thymic epithelial cell development and survival. *Cell Death Differ* **28**, 2946-2956 (2021).
8. Chaves-Moreira D, Morin PJ, Drapkin R. Unraveling the Mysteries of PAX8 in Reproductive Tract Cancers. *Cancer Res* **81**, 806-810 (2021).
9. Das S, Anczukow O, Akerman M, Krainer AR. Oncogenic splicing factor SRSF1 is a critical transcriptional target of MYC. *Cell Rep* **1**, 110-117 (2012).
10. Edison N, *et al.* Degradation of Bcl-2 by XIAP and ARTS Promotes Apoptosis. *Cell Rep* **21**, 442-454 (2017).
11. Warren CFA, Wong-Brown MW, Bowden NA. BCL-2 family isoforms in apoptosis and cancer. *Cell Death Dis* **10**, 177 (2019).
12. Stegh AH, *et al.* Glioma oncoprotein Bcl2L12 inhibits the p53 tumor suppressor. *Genes Dev* **24**, 2194-2204 (2010).
13. Stegh AH, *et al.* Bcl2L12-mediated inhibition of effector caspase-3 and caspase-7 via distinct mechanisms in glioblastoma. *Proceedings of the National Academy of Sciences* **105**, 10703-10708 (2008).

14. Kumthekar P, *et al.* A first-in-human phase 0 clinical study of RNA interference-based spherical nucleic acids in patients with recurrent glioblastoma. *Sci Transl Med* **13**, (2021).
15. Hanahan D. Hallmarks of Cancer: New Dimensions. *Cancer Discov* **12**, 31-46 (2022).
16. Krishnamurti U, Silverman JF. HER2 in breast cancer: a review and update. *Adv Anat Pathol* **21**, 100-107 (2014).
17. Oh DY, Bang YJ. HER2-targeted therapies - a role beyond breast cancer. *Nat Rev Clin Oncol* **17**, 33-48 (2020).
18. Yang MC, *et al.* Bcl2L12 with a BH3-like domain in regulating apoptosis and TMZ-induced autophagy: a prospective combination of ABT-737 and TMZ for treating glioma. *Int J Oncol* **46**, 1304-1316 (2015).
19. Stegh AH, *et al.* Bcl2L12-mediated inhibition of effector caspase-3 and caspase-7 via distinct mechanisms in glioblastoma. *Proc Natl Acad Sci U S A* **105**, 10703-10708 (2008).

REVIEWERS' COMMENTS

Reviewer #1 (Remarks to the Author):

The authors have performed many additional experiments to address the concerns I raised. The manuscript is clearly improved. Nevertheless, my specific point 2 remains as a major concern. My comment was regarding the potential issue of therapeutic index by targeting BUD31, which is a commonly essential gene basically killing almost all kinds of cells, regardless of cell origin or transformation status. After this concern can be reasonably addressed, the manuscript can be recommended for publication.

Reviewer #2 (Remarks to the Author):

The authors extensively addressed most of the concerns we suggested and improved the scientific quality of the impressive work. I have no further comments on the revised manuscript.

Reviewer #3 (Remarks to the Author):

Most of my concerns have been addressed. I still have some minor concerns:

1. I think the oncogenic role of BUD31 only partially rely on BCL2L13. Because: 1) BUD31 also promotes proliferation; 2) BCL2L13 has only reported and experimentally verified by the authors to have the function of inhibiting apoptosis.
2. line 114-116, "the expression of apoptosis-related proteins was next measured by western blot, and inactivation of BUD31 resulted in an increased Bax/Bcl-2 ratio along with increased cleaved caspase-3 and PARP1, whereas overexpression of BUD31 had opposite effects (Figures 2F, S2D and S2E)". I will suggest that the authors do not include Bax/Bcl-2 ratio as one of the apoptosis markers, as flow cytometry apoptosis assay and cleaved caspase-3 and PARP already suggested that apoptosis is induced. The authors should also add a sentence to discuss why the BCL2 level goes down after BUD31 inactivation.
3. line 343, "BCL2L12 blocks apoptosis by neutralizing effector caspase-3 and caspase-7 maturation". BCL2L12 has been reported to inhibit caspase-3 and caspase-7 through different mechanisms, but not just neutralizing effector caspase-3 and caspase-7.
4. The molecular weight should be labelled on western blots.

Response to reviewers' comments

We thank the reviewers again for your thoughtful comments on our revised manuscript. Our point-by-point responses to the reviewers' comments are included below. Text changes in the manuscript are highlighted in red.

Response to Reviewer #1 (Remarks to the Author)

----- Reviewer comments:

The authors have performed many additional experiments to address the concerns I raised. The manuscript is clearly improved. Nevertheless, my specific point 2 remains as a major concern. My comment was regarding the potential issue of therapeutic index by targeting BUD31, which is a commonly essential gene basically killing almost all kinds of cells, regardless of cell origin or transformation status. After this concern can be reasonably addressed, the manuscript can be recommended for publication.

Response: We thank the reviewer for evaluating the revised manuscript. To address the reviewer's comment, we analyzed the BUD31 expression in ovarian cancer and normal tissues from the TCGA and GTEx databases and found that BUD31 was expressed at low levels in most normal adult tissues including fallopian tube, ovary, heart, liver, and kidney, while it was expressed at high levels in large proportion of ovarian cancer tissues (**Figure A**). This finding suggests that BUD31 may be less relied upon in normal tissues than in ovarian cancer cells, and therefore inhibiting BUD31 may have less detrimental effect on normal cells than on cancer cells.

Other splicing factors have also been found to be upregulated in cancers compared with the corresponding normal tissues (Lee SC, et al. Nat Med. 2016)¹. Moreover, RNA-seq data analysis on mouse kidney in different development stages (another project of our group) revealed that most splicing factors including *Bud31* were highly expressed at early development stages and were significantly downregulated at postnatal day 42 (**Figure B**). This low expression of BUD31 in normal adult tissues

also suggest that the deleterious effect of BUD31 targeted therapy on normal tissues should be less severe than on cancer cells.

Regardless of the specificity of strategies aimed at targeting BUD31 in cancer, we would like to point out that antisense-oligonucleotides can be employed to target the alternative splice site of cancer-associated splice variants downstream of BUD31.

Indeed, our results revealed that antisense-oligonucleotide targeting BCL2L12 exon 3 showed a remarkable antitumor activity in ovarian cancer.

Response to Reviewer #2 (Remarks to the Author)

----- **Reviewer comments:**

The authors extensively addressed most of the concerns we suggested and improved the scientific quality of the impressive work. I have no further comments on the revised manuscript.

Response: We appreciate your positive comments on our revised manuscript.

Response to Reviewer #3 (Remarks to the Author)

----- **Reviewer comments:**

Most of my concerns have been addressed. I still have some minor concerns:

Response: We thank the reviewer for accepting our revisions as sufficient.

Minor points.

1. I think the oncogenic role of BUD31 only partially rely on BCL2L13. Because: 1) BUD31 also promotes proliferation; 2) BCL2L13 has only reported and experimentally verified by the authors to have the function of inhibiting apoptosis.

Response: The reviewer probably meant BCL2L12 instead of BCL2L13. We fully agree with the reviewer that the oncogenic role of BUD31 only partially rely on BCL2L12. BUD31 has other targets (E2F4 and CDK16) that are involved in proliferation of ovarian cancer cells (Revised Figure 5j-5n). We have changed the description in the manuscript.

2. line 114-116, “the expression of apoptosis-related proteins was next measured by western blot, and inactivation of BUD31 resulted in an increased Bax/Bcl-2 ratio along with increased cleaved caspase-3 and PARP1, whereas overexpression of BUD31 had opposite effects (Figures 2F, S2D and S2E)” . I will suggest that the authors do not include Bax/Bcl-2 ratio as one of the apoptosis markers, as flow cytometry apoptosis assay and cleaved caspase-3 and PARP already suggested that apoptosis is induced. The authors should also add a sentence to discuss why the BCL2 level goes down after BUD31 inactivation.

Response: Thank you for the suggestion. We have deleted “an increased Bax/Bcl-2 ratio along with” in line 114-116 and have added a sentence to discuss why the BCL2 level goes down after BUD31 inactivation.

3. line 343, “BCL2L12 blocks apoptosis by neutralizing effector caspase-3 and caspase-7 maturation”. BCL2L12 has been reported to inhibit caspase-3 and caspase-7 through different mechanisms, but not just neutralizing effector caspase-3 and caspase-

Response: We thank the reviewer for pointing this out. We have rephrased this description in the text.

4. The molecular weight should be labelled on western blots.

Response: As suggested by the reviewer, we have labeled molecular weight on western blots.

References

1. Lee SC, Abdel-Wahab O. Therapeutic targeting of splicing in cancer. *Nat Med* **22**, 976-986 (2016).